# GEOILP: A SYNTHETIC DATASET TO GUIDE LARGE-SCALE RULE INDUCTION

**Si Chen** [1]**, Richong Zhang** [1,2]**, Xu Zhang** [3] [*]
[1] SKLCCSE, Beihang University, Beijing, China
[2] Zhongguancun Laboratory, Beijing, China
[3] The National Computer Network Emergency Response Technical Team / Coordination Center of China (CNCERT/CC)
`chen.si@buaa.edu.cn, zhangrc@act.buaa.edu.cn, zhangxu@cert.org.cn`

## ABSTRACT

Inductive logic programming (ILP) is a machine learning approach aiming to learn explanatory rules from data. While existing ILP systems can successfully solve small-scale tasks, large-scale applications with various language biases are rarely explored. Besides, it is crucial for a large majority of current ILP systems to require expert-defined language bias, which hampers the development of ILP towards broader utilizations. In this paper, we introduce GeoILP, a large-scale synthetic dataset of diverse ILP tasks involving numerous aspects of language bias. These tasks are built from geometry problems, at the level from textbook exercise to regional International Mathematical Olympiad (IMO), with the help of a deduction engine. These problems are elaborately selected to cover all challenging language biases, such as recursion, predicate invention, and high arity. Experimental results show that no existing method can solve GeoILP tasks. In addition, along with classic symbolic-form data, we provide image-form data to boost the development of the joint learning of neural perception and symbolic rule induction.

## 1 INTRODUCTION

Inductive logic programming (ILP), at the intersection of machine learning (ML) and symbolic artificial intelligence, learns hypotheses from background knowledge and examples (Muggleton & De Raedt, 1994; Muggleton et al., 2012; Cropper et al., 2020a; Cropper & Dumančić, 2022; Zhang et al., 2023). ILP adopts logical formulae to represent knowledge, examples, and hypotheses uniformly. The most fascinating merit of ILP, differing from other ML approaches, is the ability to learn highly interpretable hypotheses, which reveals a potential way toward human-comprehensible, controllable, and trust-worthy artificial intelligence.

Classic symbolic ILP are based on discrete search, suffering from the combinatorially growing search space and thus restricting to small-scale scenes. To alleviate this obstacle, symbolic methods require user-defined language bias to limit searching, which is markedly crucial for efficiency (Cropper & Dumančić, 2022). Such hand-crafted work is more or less the same as the feature engineering (Khalid et al., 2014) in other ML tasks, requiring certain expert knowledge and considerably many troublesome trial and error. However, in the modern ML community, feature engineering is usually superseded by automatic feature extractors, such as various neural networks, which achieve amazing success in large-scale applications (e.g., GPT-4 (Achiam et al., 2023)). Consequently, we argue that thoroughly turning hand-crafted determination of language bias into automatic language bias discovery is a promising direction towards broader applications of ILP.

Modern neural-symbolic ILP relaxes the hypotheses space into a continuous space and leverage gradient-based optimization techniques to induce solutions, from which interpretable rules can be extracted. Despite not requiring an elaborated language bias, existing neural-symbolic methods are limited to a relatively small hypotheses space, presuming low-arity predicates, function-free clauses,

---

[*]Corresponding author

and few rule's body atoms (Glanois et al., 2022). Scaling up to large-scale scenarios is also one of the major challenges of this line of work.

However, large-scale ILP datasets are lacking in evaluating more powerful methods and guiding enhancement. Existing datasets are small or lack reference hypotheses. Our goal is to construct a large-scale ILP dataset, providing reference hypotheses to guide the resolution of present limitations in ILP, which would lead ILP to an expert-free learning paradigm (like modern deep learning) and exceedingly broader utilization. Furthermore, we aim to evaluate ILP systems without much expert priors (like other modern ML paradigms), i.e., training & testing without excessive user-defined bias.

Therefore, we construct GeoILP [1], a large-scale dataset synthesized from *plane geometry* rules that help generate reference hypotheses involving various language biases. We first adopt a symbolic deduction engine to obtain target examples from the rules and determine the background knowledge and hypotheses by tracing back from the examples. We also consider the noisy and multi-task settings, which are closer to real-world applications and actively studied in other ML tasks.

In summary, GeoILP contains 835 single-tasks and 207 multi-tasks. The predicate arity is up to 8 and the number of body atoms is up to 9. Overall, 85% single-tasks and 50% multi-tasks contain a hypothesis with the number of rules ranging from at least 10 to 100, while the rest of the tasks involve at least hundreds to thousands of rules (refer to section 5.3 for details). Besides, the language biases also involve argument symmetry, constraints, different types of recursion, and predicate invention. We conduct experiments on applicable methods, showing that GeoILP is completely unreachable.

In addition, GeoILP provides image-form background knowledge, which requires jointly training a perception network, transforming the raw sensory input (image) into symbolic knowledge, and an ILP system inducing the hypothesis. The breakthrough for such joint learning, which remains less explored, would be a breakthrough for the whole artificial intelligence community (Cropper & Dumančić, 2022).

## 2 RELATED WORK

### 2.1 ILP METHODS

**Symbolic methods** search in the hypotheses space defined by language biases. Among these, notable methods include FOIL (Quinlan, 1990), Progol (Muggleton, 1995; Muggleton & Bryant, 2000), TILED (Blockeel & De Raedt, 1997), ALEPH (SRINIVASAN, 2001), Metagol (Muggleton et al., 2015), ILASP (Law et al., 2018; 2020). These methods suffer from combinatorially growing hypotheses, noisy data, and inefficient predicate invention. Popper (Cropper & Morel, 2021a) is a modern symbolic ILP system, which is, to the best of our knowledge, the only symbolic system capable of simultaneously learning recursive rules, inventing predicates (Cropper & Morel, 2021b), handling noise (Hocquette et al., 2024), and scaling better, though still very expensive to do these.

**Neural-symbolic methods** or differentiable methods, make continuous relaxation of the discrete hypotheses space and induce solutions by minimizing loss function via gradient-based optimizer. While the early-stage methods require user-defined language templates task-by-task to restrict hypotheses (Rocktäschel & Riedel, 2017; Campero et al., 2018), the following works tend to automatically deal with more general language biases (Evans & Grefenstette, 2018; Si et al., 2019; Glanois et al., 2022). As learning interpretable solutions is the outstanding property of ILP, the methods that cannot produce human-readable rules are out of the scope of this paper (e.g., Dong et al. (2019)).

### 2.2 ILP DATASETS

**Real-world datasets** Real-world datasets collect background knowledge and examples from real-world observation. The application scenarios cover knowledge base completion (Bordes et al., 2013; Toutanova & Chen, 2015; Yang et al., 2017; Hudson & Manning, 2019), drug design (Inoue et al., 2013; Tamaddoni-Nezhad et al., 2006), ecology (Bohan et al., 2017), etc. The main demerit of

---

[1]Data is available at `https://github.com/chensi99/GeoILP`.

real-world datasets is lacking reference hypotheses. Consequently, an ILP system failing on these datasets would have little idea about where to improve.

**Synthetic datasets**   Synthetic datasets covering mathematical formal systems (Evans & Grefenstette, 2018), grammar learning (Muggleton et al., 2014; Law et al., 2019), games (Cropper et al., 2020b), program analysis (Sivaraman et al., 2019; Bartha & Cheney, 2020), etc. They can provide reference hypotheses to guide resolving the limitations of ILP systems. However, current synthetic datasets are small-scale, whose hypotheses typically contain less than 10 rules, and have already been solved by existing ILP. Our work extends this line of work to much larger scenarios.

## 3  BACKGROUND

We first introduce necessary logic notions and then define inductive logic programming (ILP). Further terminology is illustrated in the next section as well.

### 3.1  LOGIC PRELIMINARIES

**Horn clause**   Every formula in first-order language can be transformed into its semantically equivalent *conjunctive normal form*, a conjunction of clauses. A *clause* is a disjunction of literals. A *literal* is an atom (*positive literal*) or its negation (*negative literal*). An atom is called *ground* if it contains no variable. *Horn clause* is a widely used subset of clauses that allow at most one positive literal. Horn clause involve *facts*, which are atoms, and *rules* that can be semantically equivalently represented as (assumed function-free here)

$$\mathrm{H}(\boldsymbol{X}) \leftarrow \mathrm{B}_1(\boldsymbol{X}) \wedge \mathrm{B}_2(\boldsymbol{X}) \wedge \cdots \wedge \mathrm{B}_k(\boldsymbol{X})$$

where $\boldsymbol{X}$ denotes a vector of variables, the atom $\mathrm{H}(\boldsymbol{X})$ is the *head* atom of the rule, and the atoms $\mathrm{B}_1(\boldsymbol{X}), \mathrm{B}_2(\boldsymbol{X}), \dots, \mathrm{B}_k(\boldsymbol{X})$ are the *body* atoms of the rule. A *program* is a set of Horn clauses. We define *rule size* as the number of atoms in a rule and *program size* as the sum of rule size.

Note that the variables in a clause are implicitly quantified by universal quantifiers that are supposed to be placed at the beginning. The variables appearing only in the body but not the head are called *existentially quantified*.

**Forward chaining**   *Forward chaining* can be used to deduce all the true ground facts from given rules and background facts. Formally, given a set of ground atoms $\mathcal{A}$ and a Horn rule set $\mathcal{R}$, the immediate consequence through one-step forward chaining is defined as the set

$$\mathrm{con}_{\mathcal{R}}(\mathcal{A}) = \mathcal{A} \cup \left\{ \alpha \ \middle|\ \alpha \leftarrow \alpha_1, \dots, \alpha_k \in \mathrm{ground}(\mathcal{R}), \bigwedge_{i=1}^{k} \alpha_i \in \mathcal{A} \right\}$$

where $\mathrm{ground}(\mathcal{R})$ consists of all the *ground rules* instantiated from $\mathcal{R}$. Then, we recursively define the consequence through $t$ steps $\mathrm{C}_{\mathcal{R},t}(\mathcal{A})$

$$\mathrm{C}_{\mathcal{R},0}(\mathcal{A}) = \mathcal{A}, \quad \mathrm{C}_{\mathcal{R},t+1}(\mathcal{A}) = \mathrm{con}_{\mathcal{R}}\big(\mathrm{C}_{\mathcal{R},t}(\mathcal{A})\big)$$

We say the *fix point* is reached at step $T$ if $T$ is the smallest natural number satisfying $\mathrm{C}_{\mathcal{R},T}(\mathcal{A}) = \mathrm{C}_{\mathcal{R},T+1}(\mathcal{A})$, and $\mathrm{C}_{\mathcal{R},T}(\mathcal{A})$ is the set of all the consequences of the forward chaining.

### 3.2  INDUCTIVE LOGIC PROGRAMMING

We adopt the most popular ILP setting *learning from entailment* (LFE) (Cropper & Dumančić, 2022). The training data is a tuple $(\mathcal{B}, \mathcal{E}^+, \mathcal{E}^-)$ of *background knowledge* $\mathcal{B}$, *positive examples* of the concept $\mathcal{E}^+$, and *negative examples* of the concept $\mathcal{E}^-$. $\mathcal{E}^+, \mathcal{E}^-$ are sets of ground atoms relevant to the target predicate we want to learn. $\mathcal{B}$ is a set of clauses that act as background knowledge (BK), typically a set of ground atoms irrelevant to the target predicate (rules can also be in BK). The goal of ILP is to induce a *hypothesis* $\mathcal{H}$, consisting also of clauses, satisfying the following conditions

$$\forall e \in \mathcal{E}^+, \mathcal{H} \cup \mathcal{B} \models e \quad \text{(completeness)}$$
$$\forall e \in \mathcal{E}^-, \mathcal{H} \cup \mathcal{B} \nvDash e \quad \text{(consistency)}$$

The *completeness* condition states that the hypothesis and BK entail all positive examples. The *consistency* condition states that the hypothesis and BK do not entail any negative examples.

For example, given $\mathcal{B} = \{\text{Father}(\text{John}, \text{Mary}), \text{Father}(\text{Tom}, \text{John})\}$, $\mathcal{E}^+ = \{\text{Grandfather}(\text{Tom}, \text{Mary})\}$, $\mathcal{E}^- = \{\text{Grandfather}(\text{Mary}, \text{Tom}), \text{Grandfather}(\text{John}, \text{Mary})\}$, an ILP task learner may learn the following hypothesis $\mathcal{H}$ (upper italic letters denote variables)

$$\text{Grandfather}(X, Y) \leftarrow \text{Father}(X, Z) \wedge \text{Father}(Z, Y)$$

With the learned hypothesis, an automated theorem prover can derive all the facts regarding the *target predicate* Grandfather from the background facts regarding the relation Father.

## 4 LIMITATIONS OF CURRENT ILP

In this section, we identify the critical limitations of current ILP that impede the development of broader applications. Our proposed dataset is intended to cover all these challenges and is thus a good testbed for elaborating more sophisticated rule induction systems.

### 4.1 HAND-CRAFTED LANGUAGE BIAS

Language bias is used to limit the hypothesis space in symbolic ILP. As calculated by Cropper & Morel (2021a), the number of possible hypotheses grows combinatorially fast.

Without carefully human-determined language bias, such as the predicates allowed to appear in the rule's head, the predicates allowed to appear in the rule's body, enabling recursion or not, enabling predicate invention or not, the maximum number of clauses allowed in a hypothesis, the maximum number of unique variables in a clause, the maximum number of body atoms in a clause, the number of allowed existentially quantified variables, the maximum times a predicate can appear in a rule, symbolic ILP tends to be extremely slow, even useless (Cropper & Dumančić, 2022). Determining a good language bias is onerous and requires a vast amount of trial and error.

Below, we introduce the most dominant language biases, which notably increase hypothesis space and should thus be completely automatically determined by ILP systems.

**Predicate arity**   Real-world relations may involve several entities. For instance, the triadic relation $\text{Sell}(\text{seller}, \text{buyer}, \text{book})$ asserts that, in a transaction order, a seller sells a book to a buyer. However, current neuro-symbolic methods typically support arity lower than two (Evans & Grefenstette, 2018; Campero et al., 2018; Glanois et al., 2022). Several symbolic methods can support arbitrary arity but exceedingly increase search complexity (Cropper & Morel, 2021a).

**Argument symmetry**   Argument symmetry may exist for predicates. For example, if John is Mary's cousin, then Mary must also be John's cousin. As this example, argument symmetry can be represented as a Horn rule, yet complex symmetry may yield too many rules. For instance, the triadic atom asserting whether 3 people queue in a straight line evaluates to the same truth value if permuting all 3 arguments (any 3 people), which yields $3!(3! - 1) = 30$ Horn rules. To the best of our knowledge, there is no specialized way to learn compact representations for argument symmetry.

**Predicate constraint**   Atoms' truth value may be forced to be opposite under some constraints. Asymmetry constraint can be considered as Horn *goal* (clause with only negative literals) $\leftarrow \text{Pred}(X, Y) \wedge \text{Pred}(Y, X)$. The representation of other constraints (e.g., irreflexivity, anti-transitivity, anti-triangularity, functionality, exclusivity) can be found in Cropper & Hocquette (2023). Current neuro-symbolic methods do not cover this aspect.

**Recursion**   There are two types of recursion in Horn programs: recursion and mutual recursion (Bancilhon & Ramakrishnan, 1986). *Recursion* refers to the phenomenon that the same predicate appears simultaneously in a rule's head and body. For instance, $\text{Even}(X) \leftarrow \text{Even}(Y) \wedge \text{Succ}_2(Y, X)$ is recursive, where Even asserts whether a natural number is even and $\text{Succ}_2(Y, X)$ asserts whether $X = Y + 2$. Even is the *recursive predicate* in this case. Besides, this recursive rule is also called *linear* because the recursive predicate only appears once in the body. *Mutual recursion* refers to the

phenomenon that two predicates mutually derive from each other. We say a predicate $\text{Pred}_1$ *derives* another predicate $\text{Pred}_2$ if there exists such a set of rules (variables omitted)

$$\text{Pred}_2 \leftarrow \cdots \wedge \text{Q}_1 \qquad \text{Q}_1 \leftarrow \cdots \wedge \text{Q}_2 \qquad \text{Q}_2 \leftarrow \cdots \wedge \text{Q}_3 \qquad \cdots \qquad \text{Q}_n \leftarrow \cdots \wedge \text{Pred}_1$$

where $Q$s denote other predicates and $\ldots$ denote any other body atoms. Therefore, two mutually recursive predicates can be mutually deduced via forward chaining. For instance,

$$\text{Even}(X) \leftarrow \text{Odd}(Y) \wedge \text{Succ}(Y, X) \qquad \text{Odd}(X) \leftarrow \text{Even}(Y) \wedge \text{Succ}(Y, X)$$

, where $\text{Succ}(X, Y)$ asserts whether $Y = X + 1$, show that $\text{Even}$ and $\text{Odd}$ (asserting whether a natural number is odd) are mutually recursive. A rule is also called *recursive* if the head predicate is mutually recursive with one of its body predicates, and the rule is called *linear* if only one mutually recursive predicate appears in the body. Enabling recursion is expensive for symbolic ILP, while mutual recursion between any two predicates is not supported by the state-of-the-art neuro-symbolic ILP (Glanois et al., 2022).

**Predicate invention** Predicate invention is a crucial part of automatically discovering new concepts, which may lead to breakthroughs in AI development (Russell, 2019, chap. 3). Specifically, predicate invention enables predicates that are unused in BK & target examples appearing in the hypothesis. For example, learning $\text{Even}$ from the BK $\{\text{Zero}(0), \text{Succ}(0, 1), \text{Succ}(1, 2), \text{Succ}(2, 3), \ldots\}$ may require inventing dyadic relation $\text{Succ}_2$ and the following rules

$$\text{Even}(X) \leftarrow \text{Even}(Y) \wedge \text{Succ}_2(Y, X) \qquad \text{Even}(0) \leftarrow \text{Zero}(0)$$
$$\text{Succ}_2(Y, X) \leftarrow \text{Succ}(Z, X) \wedge \text{Succ}(Y, Z)$$

While inventing such auxiliary predicates substantially reduces hypotheses (Cropper & Dumančić, 2022) and improves learning performance (Cropper, 2019), predicate invention is expensive, inaccurate, and restricted to low-arity invention (Cropper & Morel, 2021b).

## 4.2 INSUFFICIENT NOISE HANDLING

**Mislabeled & ambiguous data** Noise is ubiquitous in realistic data. While symbolic methods struggle to learn from noisy data (Hocquette et al., 2024), neuro-symbolic methods can deal with mislabeled examples (Glanois et al., 2022) and ambiguous BK (Evans & Grefenstette, 2018). Handling mislabeled BK is still an open problem (Cropper & Dumančić, 2022).

**Open-world assumption** While the *closed-world assumption* (CWA) asserts any ground atom, whose predicate appears in the BK, to be false if it is not given in BK, the *open-world assumption* (OWA) allows those ground atoms that are not known to be true to have the possibility of being true (Reiter, 1981). Almost all existing ILP systems assume CWA. However, OWA is a more realistic setup since a complete BK is inaccessible in real-world applications. A set of incomplete background ground atoms is considered noisy if an ILP system assumes CWA.

## 4.3 MULTI-TASK LEARNING

Existing ILP focuses on once learning one target predicate. However, simultaneously learning several target predicates can share common rules and capture mutual recursions among target predicates. Glanois et al. (2022) proposes an iterative multi-task learning scheme for their neuro-symbolic model and successfully learns certain hypotheses at a small program size. Large-scale multi-task learning is a promising direction for building broader applications, which remains unexplored.

## 4.4 UNABLE TO LEARN FROM RAW INPUT

One major gap between ILP and modern ML systems is the ability to induce knowledge from raw sensory input. Most ILP systems only receive symbolic data as input, while raw data is usually images, speech, natural language, etc. There are initial works that use neural networks to perceive and transform raw input into symbolic form and use symbolic deduction to do reasoning (Manhaeve et al., 2018; Dai et al., 2019; Dai & Muggleton, 2021). Efficiently jointly training perception network and ILP system in large-scale tasks would be a promising direction for robust and reliable artificial intelligence. See Evans (2020); Evans et al. (2021; 2022) for more possibilities to induce rules from raw sensory input.

Table 1: Predicates used in GeoILP (Target=can be used as target predicate; Head=can be used in the head; Body=can be used in the body). Details see Appendix A.1.

| Name | Arity | Target | Argument Symmetry | Constraints | Head | Body | Full Name |
|------|-------|--------|-------------------|-------------|------|------|-----------|
| Coll | 3 | ✗ | ✓ | ✓ | ✓ | ✓ | collinear |
| Ncoll | 3 | ✗ | ✓ | ✓ | ✓ | ✓ | not collinear |
| Sameside | 6 | ✗ | ✓ | ✓ | ✗ | ✓ | same side |
| Midp | 3 | ✓ | ✓ | ✓ | ✓ | ✓ | midpoint |
| Cong | 4 | ✓ | ✓ | ✓ | ✓ | ✓ | segment congruent |
| Perp | 4 | ✓ | ✓ | ✓ | ✓ | ✓ | perpendicular |
| Para | 4 | ✓ | ✓ | ✓ | ✓ | ✓ | parallel |
| Cyclic | 4 | ✓ | ✓ | ✓ | ✓ | ✓ | concyclic |
| Circle | 4 | ✓ | ✓ | ✓ | ✓ | ✓ | circle |
| Eqangle | 8 | ✓ | ✓ | ✓ | ✓ | ✓ | equal angle |
| Eqratio | 8 | ✓ | ✓ | ✓ | ✓ | ✓ | equal ratio |

## 5 GEOILP

In this section, we formally introduce GeoILP, our proposed dataset for elaborating large-scale sophisticated rule induction systems, in four parts. First, a general guide for synthesizing ILP tasks from predefined rules. Second, the steps for identifying the examples (*deduction* step) and the BK (*traceback* step) of GeoILP. Third, the features of GeoILP critically differ from other ILP testbeds. Fourth, the approach to transform symbolic input into raw sensory input.

### 5.1 A GENERAL GUIDE FOR ILP TASK SYNTHESIS

Inductive reasoning can be seen as a "reverse" procedure of deductive reasoning in the sense that the former learns rules from premises and conclusions while the latter infers conclusions from premises and valid rules. In ILP, premises correspond to the BK, and the conclusions correspond to the target examples. Therefore, to construct ILP data, we can derive target examples from selected premises and predefined rules using a deduction engine.

Concretely, the data-synthesizing procedure works as follows:

1. Randomly or intentionally choose several ground atoms as premises.
2. Define a consistent set of rules. [2]
3. Deduce all the conclusions from the premises and rules using any deduction engine.
4. Select a part or all of the conclusions (also ground atoms) as **positive examples**. [3]
5. Trace back from the conclusions to identify a minimum set of premises contributing to deducing the conclusions as ILP **BK**.
6. Optionally, obtain **negative examples** by removing the conclusions from all syntactically possible ground atoms (all combinations of target predicate and constants).

### 5.2 SYNTHESIZING GEOILP

We choose *plane geometry* as the application domain as its formal system covers all the difficulties indicated in section 4 (see section 5.3).

Formalizing plane geometry and building a corresponding symbolic deduction engine are challenging works outside this paper's scope. We adopt an expert-designed deduction engine based on deductive database theory (Gallaire et al., 1984), similar to the ones used in automated geometry

---

[2]A set of rules is *consistent* if the rules do not contradict each other.

[3]Merely selecting a part of the conclusions as positive examples may yield fewer rules with more body atoms. $P_1 \leftarrow P_2 \wedge Q$ and $P_2 \leftarrow R_1 \wedge R_2$ may be reduced to $P_1 \leftarrow R_1 \wedge R_2 \wedge Q$ if the premises involve $Q, R_1, R_2$ and only select $P_1$ as conclusions.

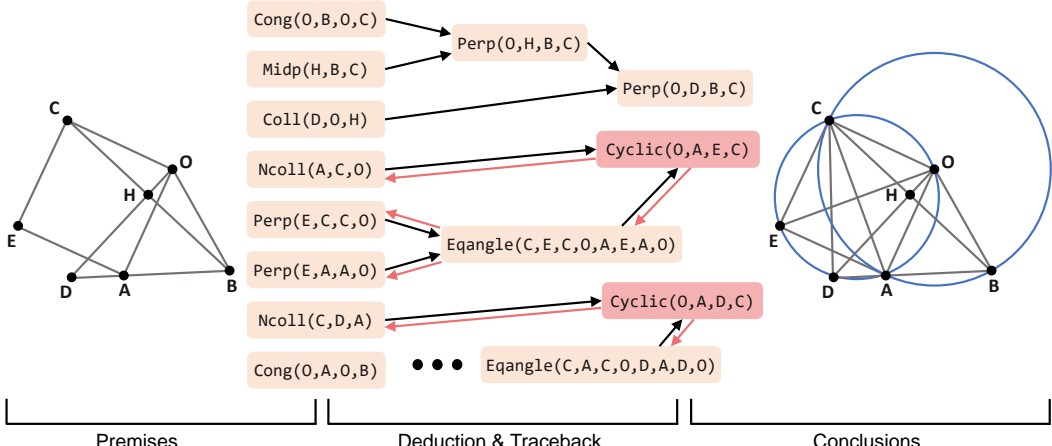

Figure 1: Synthesizing one of GeoILP (single) tasks from one set of premises with Cyclic as target predicate. Black arrows denote deduction (forward chaining), and red arrows denote traceback.

theorem proving (e.g., GEX (Chou et al., 2000), JGEX (Ye et al., 2010a;b; 2011), AlphaGeometry (Trinh et al., 2024)). [4] The only constants are the points in the plane. The engine leverages a set of Horn rules for deduction. Table 1 shows the characteristics of predicates.

**Deduction**   To initialize *deduction* step with premises, we use plane geometry problems given in JGEX (Ye et al., 2010a;b; 2011), ranging from textbook exercises, regional olympiads, and famous theorems. An example of premises is depicted in Figure 1. The final dataset, built from such selected premises, can effectively help construct automated geometry theorem provers without needing expert-defined rules, as the rules learned by ILP can be useful. Then, the deduction engine uses forward chaining to reach the fix point. In the single-task setting, we separate conclusions with different predicates.

Note that the deduction engine regards argument-permutation equivalent atoms as the same, which substantially reduces deduction costs since argument symmetry is omnipresent. Therefore, rules for argument permutation and several trivial rules are not explicitly listed. See Appendix A.4 for details.

**Traceback**   Since the conclusions deduced from one set of premises may involve all 8 predicates, synthesizing a single ILP task should filter out those premises irrelevant to the target predicate. To achieve this, the deduction engine constructs a deduction graph when doing forward chaining, which illustrates the immediate dependence of the ground atoms in the graph. Every body atom in a matched (ground) rule has a directed edge pointing to the head atom (see Figure 1). Starting from the conclusions involving only the target predicate, we trace back along the directed edges in the reverse direction until reaching the premises. The trace-backed premises are regarded as the BK. The directed edges alongside (red arrows in Figure 1) constitute a reference hypothesis.

After deduction and traceback, we repeat the BK and target examples ten times, retaining predicates unchanged but mapping every point to new, unique points. In other words, the initial group of points is duplicated into ten groups. Then, the data are divided into training set and evaluation set according to 8:2 point groups.

## 5.3   DATASET FEATURES

In total, 63 expert-defined rules and many more trivial rules encoded in the deductive database are used for synthesizing GeoILP (see Appendix A.4). Several rules are listed here to demonstrate how

---

[4]Note that GEX, JGEX, and AlphaGeometry are deductive reasoning algorithms and are not applicable to ILP, which is inductive reasoning.

Table 2: Detailed comparison between GeoILP and the existing dataset. (# denotes *number of*)

| Dataset | $\partial$**ILP** | **GeoILP** |
|---|---|---|
| #Tasks | 20 single-tasks | 835 single-tasks; 207 multi-tasks |
| Predicate Arity | 1 - 2 | 3 - 8 |
| Predicate Invention | $\sim 60\%$ | 96% for single-tasks |
| Max #Variables | 6 | 12 |
| Max #Body Atoms | 2 | 9 |
| #Rules | 1 - 5 | 10-100 for 85% single-tasks and 50% multi-tasks; $\leq 1783$ for single-tasks; $\leq 3553$ for multi-tasks |
| Argument Symmetry | in just a few predicates; limited to dyadic predicates | in every predicate; various symmetries in up to ogdoadic predicates |
| Predicate Constraints | in a few predicates | in every predicate |
| Recursion | linear recursion: 30% non-linear recursion: 10% linear mutual recursion: 10% | recursion: $\sim 100\%$ mutual recursion: $\sim 90\%$ all 4 kinds of recursion are common |

GeoILP covers the limitations mentioned in section 4.

$$\langle 1 \rangle \qquad \mathrm{Midp}(M, C, D) \leftarrow \mathrm{Midp}(M, A, B) \wedge \mathrm{Para}(A, C, B, D) \wedge \mathrm{Para}(A, D, B, C)$$
$$\langle 2 \rangle \qquad \mathrm{Para}(A, B, E, F) \leftarrow \mathrm{Para}(A, B, C, D) \wedge \mathrm{Para}(C, D, E, F)$$
$$\langle 3 \rangle \qquad \mathrm{Cong}(A, M, B, M) \leftarrow \mathrm{Perp}(A, B, B, C) \wedge \mathrm{Midp}(M, A, C)$$
$$\langle 4 \rangle \qquad \mathrm{Cong}(O, A, O, B) \leftarrow \mathrm{Midp}(M, A, B) \wedge \mathrm{Perp}(O, M, A, B)$$
$$\langle 5 \rangle \qquad \mathrm{Perp}(A, B, P, Q) \leftarrow \mathrm{Cong}(A, P, B, P) \wedge \mathrm{Cong}(A, Q, B, Q)$$

We briefly introduce the meaning of the predicates appearing in the above rules and Figure 1 to facilitate understanding them. See Appendix A.1 for exhaustive descriptions. $\mathrm{Midp}(M, A, B)$ asserts $M$ is the midpoint of segment $AB$. $\mathrm{Para}(A, B, C, D)$ asserts lines $AB$ & $CD$ are parallel. $\mathrm{Perp}(A, B, C, D)$ asserts lines $AB$ & $CD$ are perpendicular. $\mathrm{Cong}(A, B, C, D)$ asserts segments $AB$ & $CD$ are of same length. $\mathrm{Coll}(A, B, C)$ asserts $A, B, C$ are collinear. $\mathrm{Ncoll}(A, B, C)$ asserts $A, B, C$ are not collinear. $\mathrm{Cyclic}(A, B, C, D)$ asserts $A, B, C, D$ are concyclic. $\mathrm{Eqangle}(A, B, C, D, E, F, G, H)$ asserts full-angles $[AB, CD]$ & $[EF, GH]$ are equal. Full-angle is defined by two lines and, intuitively, two full-angles $[AB, CD]$ & $[EF, GH]$ are equal if, supposing $Rot$ denotes a rotation, $Rot(AB) \parallel EF$ and $Rot(EF) \parallel GH$. Refer to Ye et al. (2010b) for the formal definition.

**Predicate** Characteristics of predicates are provided in Table 1. The predicates are of arity from 3 to 8, all involving argument symmetry (see Appendix A.2) and different constraints (see Appendix A.3). Note that, among constraints, $\mathrm{Midp}$ and $\mathrm{Circle}$ are functional; thus, our dataset can be easily adapted to study the setting with functions. In addition, nearly all the predicates can be used in the head and body of a rule.

**Rules** The number of rules in 85% single-tasks ranges from at least 10 to 100, while the rest of single-tasks involve hundreds to thousands of rules. These rules result from expert-defined rules, argument symmetry rules, and trivial rules. See Appendix A.4 for details about trivial rules. The maximum number of body atoms is 9. The maximum number of variables in a rule is 12. Existentially quantified variables usually exist. For instance, $A, B$ of Rule$\langle 1 \rangle$, $C, D$ of Rule$\langle 2 \rangle$, $C$ of Rule$\langle 3 \rangle$, $M$ of Rule$\langle 4 \rangle$.

**Recursion**   All types of recursion are omnipresent in the whole rule set, and almost any two predicates are mutually recursive. In the example rule subset above, Rule⟨1⟩ is recursive and linear; Rule⟨2⟩ is recursive but not linear; Rule⟨3⟩ & Rule⟨5⟩ (or Rule⟨4⟩ & Rule⟨5⟩) justify that they are recursive because Cong and Perp are mutually recursive; while Rule⟨3⟩ (also Rule⟨4⟩) is linear, Rule⟨5⟩ is not linear.

**Predicate invention**   Almost all GeoILP tasks require predicate invention. For example, a task learning Para with Cong, Coll, Ncoll, Eqangle, Perp may need to invent Midp as auxiliary predicate (Appendix B.1). In addition, extra meaningful relations that are not used in GeoILP may also be invented to reduce hypothesis space, e.g., congruent triangle, similar triangle (Appendix B.1).

**Noisy data**   First, our synthetic data makes the open-world assumption. For instance, in Figure 1, Perp appears twice in the BK (premises), while two new atoms of Perp not given in the BK also appear in the deduced conclusions, which means that atoms not given in the BK can also be true. In GeoILP, it is common for all true atoms not to be given in BK. Second, the negative examples are noisy since our rule set is incomplete for the entire *plane geometry* (i.e., several true atoms may not be deduced based on the incomplete rule set).

**Multi-task**   In the multi-task setting, we trace back premises from all conclusions; thus, all predicates in the deduction graph are considered target predicates for an ILP multi-task. [5] In most cases, the trace-backed premises are the same as the initial premises.

### 5.3.1   COMPARISON WITH OTHER ILP DATASETS

We compare GeoILP with the dataset proposed in ∂ILP (Evans & Grefenstette, 2018), which is the only synthetic dataset used by recent neuro-symbolic methods (e.g., HRI [20]). Table 2 reveals GeoILP's extremely strong complexity from various aspects. [6] [7]

### 5.4   CONSTRUCTING RAW INPUT

An essential difference distinguishing it from other datasets is that GeoILP additionally provides raw inputs corresponding to each task. Like in Figure 1, the BK (premises) is transformed into an image like in the plane geometry textbook. We adopt the constructive diagram builder language developed in AlphaGeometry (Trinh et al., 2024) to construct the image point by point from a given set of premises, which works well with the symbolic deduction engine. The goal is to provide data for learning rules from raw sub-symbolic inputs (images) and symbolic target examples. The images only contain basic geometry objects, reducing the burden of perception and making them a good testbed for this immature research topic. We also attach a corresponding image of conclusions (BK + positive examples) to each task, like the rightmost diagram in Figure 1. Handling GeoILP in geometric form at least requires the ability to identify geometric objects, to identify the relations among objects, and to induce interpretable rules. Developing such a complex system requires great effort, which is out of the scope of a dataset constructing work and is left to future work. [8]

More details about datasets and example data are provided in Appendix A & B, respectively.

## 6   EXPERIMENTS

### 6.1   SETUP

Considering the great difficulty of GeoILP, we divide it into four progressive levels, which provides chances for gradually strengthening ILP systems. Table 3 illustrates the specification of each level.

---

[5]Hence, multi-tasks do not involve predicate invention.

[6]The numbers of rules are lower bounds of actual numbers, since it is hard to count all trivial rules and argument symmetry rules that are implicitly encoded in the deduction engine.

[7]These statistics are from our provided reference hypotheses. An ILP system may learn other possible hypotheses with different statistics.

[8]For readers interested in the induction ability of large language models, we provide a guide on how to translate GeoILP into natural-language form in Appendix C.

Table 3: GeoILP's four single-task levels & multi-task level.

| Level | basic | simple | advanced | complex | multi-task |
|---|---|---|---|---|---|
| (single) tasks% | 2% | 9% | 23% | 66% | - |
| arity | $\leq 8$ | $\leq 8$ | $\leq 8$ | $\leq 8$ | $\leq 8$ |
| # body atoms | $\leq 2$ | $\leq 9$ | $\leq 9$ | $\leq 9$ | $\leq 9$ |
| mutual recursion? | ✗ | ✗ | ✓ | ✓ | ✓ |
| # rules | $\leq 20$ | $\leq 20$ | $\leq 20$ | $\leq 1783$ | $\leq 3553$ |

The four levels are set according to four dimensionalities: predicate arity, number of body atoms, involving mutual recursion or not, and number of rules. The former three are critical bottlenecks of many neuro-symbolic ILP methods, e.g., Glanois et al. (2022). The last one is a critical aspect affecting search complexity for both symbolic and neuro-symbolic ILP methods.

## 6.2 RESULTS

**Symbolic** Among symbolic methods, Popper (Cropper & Morel, 2021a) is the most powerful one that simultaneously supports learning recursion, involving hypothesis constraints, inventing predicates (Cropper & Morel, 2021b), and handling noise (Hocquette et al., 2024), and scales better as well. We conduct experiments using Popper [9], enabling predicate invention, recursion and noise handling. Noise handling is turned on because GeoILP follows OWA, while Popper follows CWA. The maximum number of variables in a rule is set to 12, which is the maximum value in every four levels. When conducting experiments on different levels, we set the maximum number of body atoms and the maximum number of rules to the maximum values of the learning level. This setup aligns with our purpose of not injecting many priors into training. After 1-day searching, Popper does not return any hypothesis, even at the *basic* level. We regard GeoILP as unsolvable by Popper since the searching time is already about two orders of magnitude longer than in previous works (about hundreds of seconds or less, e.g., Cropper & Morel (2021a); Glanois et al. (2022))

**Neuro-symbolic** Several neuro-symbolic methods (Evans & Grefenstette, 2018; Glanois et al., 2022) do not support higher-arity predicates, rules with more than two body atoms, or using target predicate in the rule's body, which are their primary bottlenecks of being inapplicable to GeoILP. Several others (Rocktäschel & Riedel, 2017; Campero et al., 2018) require expert-defined rule templates task-by-task, which is inappropriate for large-scale applications like GeoILP. Difflog (Si et al., 2019) is a neural-symbolic method that supports arbitrary hypothesis space. However, our experiments show that, even at the *basic* level, Difflog [10] throws an out-of-memory error on a server with 500GB of memory, an order of magnitude larger than in the original paper (64GB). We leave further investigation on improving memory usage for future work.

## 7 DISCUSSION AND CONCLUSION

We propose GeoILP, a large-scale synthetic dataset for inductive logic programming involving all challenging language biases in reference hypotheses. GeoILP is, in terms of the hypothesis size, at least one magnitude larger than existing datasets that can provide guiding hypotheses. Although GeoILP may be biased towards *plane geometry*, it is still a good testbed for large-scale ILP. Besides, we also provide image-form background knowledge, aiming to boost the development of joint learning of neural perception and symbolic rule induction.

---

[9]Version 4.3.0: `https://github.com/logic-and-learning-lab/Popper/tree/v4.3.0`
[10]We leverage the implementation and recommend parameter setting in `https://github.com/petablox/difflog/tree/3c2d5218d9a0a1e200ebbf2d6a1e5d077fb18826`.

ACKNOWLEDGMENTS

This work was supported by the National Natural Science Foundation of China (No. U23B2056), in part by the Fundamental Research Funds for the Central Universities, and in part by the State Key Laboratory of Complex & Critical Software Environment.

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

## A  DETAILS OF GEOILP

### A.1  PREDICATES DESCRIPTION

In this section, we exhaustively describe the predicates.

The constants in GeoILP are points in the plane. The exact meaning of each predicate is listed below.

- $\mathrm{Midp}(\mathrm{M, A, B})$ asserts $M$ is the midpoint of segment $AB$.
- $\mathrm{Para}(\mathrm{A, B, C, D})$ asserts line $AB$ and line $CD$ are parallel.
- $\mathrm{Perp}(\mathrm{A, B, C, D})$ asserts line $AB$ and line $CD$ are perpendicular.
- $\mathrm{Cong}(\mathrm{A, B, C, D})$ asserts segment $AB$ and segment $CD$ are of same length.
- $\mathrm{Coll}(\mathrm{A, B, C})$ asserts $A, B, C$ are collinear.
- $\mathrm{Ncoll}(\mathrm{A, B, C})$ asserts $A, B, C$ are not collinear.
- $\mathrm{Cyclic}(\mathrm{A, B, C, D})$ asserts $A, B, C, D$ are concyclic.
- $\mathrm{Eqangle}(\mathrm{A, B, C, D, E, F, G, H})$ asserts full-angles $[AB, CD]$ & $[EF, GH]$ are equal.
  - Full-angle is defined by two lines. Intuitively, two full-angles $[AB, CD]$ & $[EF, GH]$ are equal if, supposing $Rot$ denotes a rotation, $Rot(AB) \parallel EF$ and $Rot(CD) \parallel GH$.
  - Refer to Ye et al. (2010b) for the formal definition.
- $\mathrm{Eqratio}(\mathrm{A, B, C, D, E, F, G, H})$ asserts the ratio of segment $AB$ and segment $CD$ equals to the ratio of segment $EF$ and $GH$.
- $\mathrm{Circle}(\mathrm{O, A, B, C})$ asserts $A, B, C$ are concyclic and the center is $O$.
- $\mathrm{Sameside}(B, A, C, Y, X, Z)$ asserts $B$ is to the same side of $A\&C$ as $Y$ is to $X\&Z$. Mathematically, this is equivalent to $(\vec{BA} \cdot \vec{BC})(\vec{YX} \cdot \vec{YZ}) > 0$.

Sameside are not allowed as target predicates because it does not appear in the head of any rules. Coll and Ncoll are not allowed as target predicates because the rules where they appear in the head are either too few or trivial (see Appendix A.4 for trivial rules).

### A.2  ARGUMENT SYMMETRY OF PREDICATES

We use biconditional to describe the argument symmetry of predicates.

- $\mathrm{Midp}(\mathrm{M, A, B}) \leftrightarrow \mathrm{Midp}(M, B, A)$
- $\mathrm{Para}(\mathrm{A, B, C, D}) \leftrightarrow \mathrm{Para}(C, D, A, B)$
- $\mathrm{Para}(\mathrm{A, B, C, D}) \leftrightarrow \mathrm{Para}(B, A, C, D)$
- $\mathrm{Perp}(\mathrm{A, B, C, D}) \leftrightarrow \mathrm{Perp}(C, D, A, B)$
- $\mathrm{Perp}(\mathrm{A, B, C, D}) \leftrightarrow \mathrm{Perp}(B, A, C, D)$
- $\mathrm{Cong}(\mathrm{A, B, C, D}) \leftrightarrow \mathrm{Cong}(C, D, A, B)$
- $\mathrm{Cong}(\mathrm{A, B, C, D}) \leftrightarrow \mathrm{Cong}(B, A, C, D)$
- $\mathrm{Coll}(\mathrm{A, B, C}) \leftrightarrow \mathrm{Coll}(A, C, B)$
- $\mathrm{Coll}(\mathrm{A, B, C}) \leftrightarrow \mathrm{Coll}(B, C, A)$
- $\mathrm{Ncoll}(\mathrm{A, B, C}) \leftrightarrow \mathrm{Ncoll}(A, C, B)$
- $\mathrm{Ncoll}(\mathrm{A, B, C}) \leftrightarrow \mathrm{Ncoll}(B, C, A)$
- $\mathrm{Cyclic}(\mathrm{A, B, C, D}) \leftrightarrow \mathrm{Cyclic}(B, C, D, A)$
- $\mathrm{Cyclic}(\mathrm{A, B, C, D}) \leftrightarrow \mathrm{Cyclic}(B, A, C, D)$
- $\mathrm{Circle}(O, A, B, C) \leftrightarrow \mathrm{Circle}(O, B, C, A)$
- $\mathrm{Circle}(O, A, B, C) \leftrightarrow \mathrm{Circle}(O, A, C, B)$
- $\mathrm{Sameside}(B, A, C, Y, X, Z) \leftrightarrow \mathrm{Sameside}(Y, X, Z, B, A, C)$

- $\text{Sameside}(B, A, C, Y, X, Z) \leftrightarrow \text{Sameside}(B, C, A, Y, X, Z)$
- $\text{Eqangle}(A, B, C, D, E, F, G, H) \leftrightarrow \text{Eqangle}(E, F, G, H, A, B, C, D)$
- $\text{Eqangle}(A, B, C, D, E, F, G, H) \leftrightarrow \text{Eqangle}(C, D, A, B, G, H, E, F)$
- $\text{Eqangle}(A, B, C, D, E, F, G, H) \leftrightarrow \text{Eqangle}(B, A, C, D, E, F, G, H)$
- $\text{Eqratio}(A, B, C, D, E, F, G, H) \leftrightarrow \text{Eqratio}(E, F, G, H, A, B, C, D)$
- $\text{Eqratio}(A, B, C, D, E, F, G, H) \leftrightarrow \text{Eqratio}(C, D, A, B, G, H, E, F)$
- $\text{Eqratio}(A, B, C, D, E, F, G, H) \leftrightarrow \text{Eqratio}(B, A, C, D, E, F, G, H)$

Note that these biconditional formulae constitute a minimal representation of argument symmetry. Much more symmetries can be derived by combining several of them.

The rule set would be too large if these argument symmetries were all represented as Horn rules. A compact representation and effective handling should be thus considered by ILP methods.

### A.3    Constraints of predicates

We describe the constraints of predicates. We express the constraints by Horn goal, as in Cropper & Hocquette (2023).

- $\leftarrow \text{Sameside}(B, A, C, Y, X, Z) \wedge \text{Sameside}(A, B, C, Y, X, Z)$
- $\leftarrow \text{Midp}(M, A, B) \wedge \text{Midp}(N, A, B) \wedge M \neq N$
- $\leftarrow \text{Circle}(O, A, B, C) \wedge \text{Circle}(Q, A, B, C) \wedge O \neq Q$
- $\leftarrow \text{Perp}(A, B, C, D) \wedge \text{Perp}(C, D, E, F) \wedge \text{Perp}(A, B, E, F)$
- $\leftarrow \quad \text{Eqangle}(A, B, C, D, E, F, G, H) \quad \wedge \quad \text{Eqangle}(C, D, A, B, E, F, G, H) \quad \wedge \\ Npara(A, B, C, D)$
- $\leftarrow \quad \text{Eqangle}(A, B, C, D, E, F, G, H) \quad \wedge \quad \text{Eqangle}(C, D, A, B, E, F, G, H) \quad \wedge \\ Ncong(A, B, C, D)$

(Npara for not parallel; Ncong for not congruent)

### A.4    Rules

All the Horn rules utilized to synthesize GeoILP are divided into expert-defined rules, trivial rules, and argument-symmetry rules (if representing argument symmetries by Horn rules).

Trivial rules are Horn rules trivial for human. For example, $\text{Coll}(A, B, C) \leftarrow \text{Coll}(A, D, B) \wedge \text{Coll}(A, D, C)$. Note that, though trivial for human, an ILP learner may suffer from the large number of trivial rules. A whole set of such trivial rules is illustrated in Chou et al. (2000) and is encoded into the deductive database.

The expert-defined rules are

1. $\text{Para}(A, B, E, F) \leftarrow \text{Perp}(A, B, C, D) \wedge \text{Perp}(C, D, E, F) \wedge \text{Ncoll}(A, B, E)$
2. $\text{Cyclic}(A, B, C, D) \leftarrow \text{Cong}(O, A, O, B) \wedge \text{Cong}(O, B, O, C) \wedge \text{Cong}(O, C, O, D)$
3. $\text{Para}(A, B, C, D) \leftarrow \text{Eqangle}(A, B, P, Q, C, D, P, Q)$
4. $\text{Eqangle}(P, A, P, B, Q, A, Q, B) \leftarrow \text{Cyclic}(A, B, P, Q)$
5. $\text{Cyclic}(A, B, P, Q) \leftarrow \text{Eqangle}(P, A, P, B, Q, A, Q, B) \wedge \text{Ncoll}(P, Q, A)$
6. $\text{Cong}(A, B, P, Q) \leftarrow \text{Cyclic}(A, B, C, P) \wedge \text{Cyclic}(A, B, C, Q) \wedge \text{Cyclic}(A, B, C, R) \wedge \\ \text{Eqangle}(C, A, C, B, R, P, R, Q)$
7. $\text{Para}(E, F, B, C) \leftarrow \text{Midp}(E, A, B) \wedge \text{Midp}(F, A, C)$
8. $\text{Eqratio3}(A, B, C, D, O, O) \leftarrow \text{Para}(A, B, C, D) \wedge \text{Coll}(O, A, C) \wedge \text{Coll}(O, B, D)$
9. $\text{Eqangle}(A, B, E, F, C, D, G, H) \leftarrow \text{Perp}(A, B, C, D) \wedge \text{Perp}(E, F, G, H)$
10. $\text{Eqangle}(A, B, E, F, M, N, R, U) \quad \leftarrow \quad \text{Eqangle}(A, B, C, D, M, N, P, Q) \quad \wedge \\ \text{Eqangle}(C, D, E, F, P, Q, R, U)$

11. $\text{Eqratio}(A, B, E, F, M, N, R, U) \quad \leftarrow \quad \text{Eqratio}(A, B, C, D, M, N, P, Q) \quad \wedge$
$\text{Eqratio}(C, D, E, F, P, Q, R, U)$

12. $\text{Eqangle}(A, B, A, D, A, D, A, C) \leftarrow \text{Eqratio}(D, B, D, C, A, B, A, C) \wedge \text{Coll}(D, B, C) \wedge$
$\text{Ncoll}(A, B, C)$

13. $\text{Eqratio}(D, B, D, C, A, B, A, C) \leftarrow \text{Eqangle}(A, B, A, D, A, D, A, C) \wedge \text{Coll}(D, B, C) \wedge$
$\text{Ncoll}(A, B, C)$

14. $\text{Eqangle}(O, A, A, B, A, B, O, B) \leftarrow \text{Cong}(O, A, O, B) \wedge \text{Ncoll}(O, A, B)$

15. $\text{Cong}(O, A, O, B) \leftarrow \text{Eqangle}(A, O, A, B, B, A, B, O) \wedge \text{Ncoll}(O, A, B)$

16. $\text{Eqangle}(A, X, A, B, C, A, C, B) \leftarrow \text{Circle}(O, A, B, C) \wedge \text{Perp}(O, A, A, X)$

17. $\text{Perp}(O, A, A, X) \leftarrow \text{Circle}(O, A, B, C) \wedge \text{Eqangle}(A, X, A, B, C, A, C, B)$

18. $\text{Eqangle}(A, B, A, C, O, B, O, M) \leftarrow \text{Circle}(O, A, B, C) \wedge \text{Midp}(M, B, C)$

19. $\text{Midp}(M, B, C) \quad \leftarrow \quad \text{Circle}(O, A, B, C) \quad \wedge \quad \text{Coll}(M, B, C) \quad \wedge$
$\text{Eqangle}(A, B, A, C, O, B, O, M)$

20. $\text{Cong}(A, M, B, M) \leftarrow \text{Perp}(A, B, B, C) \wedge \text{Midp}(M, A, C)$

21. $\text{Perp}(A, B, B, C) \leftarrow \text{Circle}(O, A, B, C) \wedge \text{Coll}(O, A, C)$

22. $\text{Eqangle}(A, D, C, D, C, D, C, B) \leftarrow \text{Cyclic}(A, B, C, D) \wedge \text{Para}(A, B, C, D)$

23. $\text{Cong}(O, A, O, B) \leftarrow \text{Midp}(M, A, B) \wedge \text{Perp}(O, M, A, B)$

24. $\text{Perp}(A, B, P, Q) \leftarrow \text{Cong}(A, P, B, P) \wedge \text{Cong}(A, Q, B, Q)$

25. $\text{Perp}(P, A, A, Q) \leftarrow \text{Cong}(A, P, B, P) \wedge \text{Cong}(A, Q, B, Q) \wedge \text{Cyclic}(A, B, P, Q)$

26. $\text{Para}(A, C, B, D) \leftarrow \text{Midp}(M, A, B) \wedge \text{Midp}(M, C, D)$

27. $\text{Midp}(M, C, D) \leftarrow \text{Midp}(M, A, B) \wedge \text{Para}(A, C, B, D) \wedge \text{Para}(A, D, B, C)$

28. $\text{Para}(A, B, C, D) \leftarrow \text{Eqratio}(O, A, A, C, O, B, B, D) \wedge \text{Coll}(O, A, C) \wedge \text{Coll}(O, B, D) \wedge$
$\text{Ncoll}(A, B, C) \wedge \text{Sameside}(A, O, C, B, O, D)$

29. $\text{Coll}(A, B, C) \leftarrow \text{Para}(A, B, A, C)$

30. $\text{Eqratio}(M, A, A, B, N, C, C, D) \leftarrow \text{Midp}(M, A, B) \wedge \text{Midp}(N, C, D)$

31. $\text{Perp}(A, B, C, D) \leftarrow \text{Eqangle}(A, B, P, Q, C, D, U, V) \wedge \text{Perp}(P, Q, U, V)$

32. $\text{Cong}(A, B, C, D) \leftarrow \text{Eqratio}(A, B, P, Q, C, D, U, V) \wedge \text{Cong}(P, Q, U, V)$

33. $\text{Eqangle}(A, B, A, C, P, Q, P, R) \quad \leftarrow \quad \text{Cong}(A, B, P, Q) \wedge \text{Cong}(B, C, Q, R) \wedge$
$\text{Cong}(C, A, R, P) \wedge \text{Ncoll}(A, B, C)$

34. $\text{Eqangle}(B, A, B, C, Q, P, Q, R) \quad \leftarrow \quad \text{Cong}(A, B, P, Q) \wedge \text{Cong}(B, C, Q, R) \wedge$
$\text{Cong}(C, A, R, P) \wedge \text{Ncoll}(A, B, C)$

35. $\text{Eqangle}(C, A, C, B, R, P, R, Q) \quad \leftarrow \quad \text{Cong}(A, B, P, Q) \wedge \text{Cong}(B, C, Q, R) \wedge$
$\text{Cong}(C, A, R, P) \wedge \text{Ncoll}(A, B, C)$

36. $\text{Cong}(A, C, P, R) \quad \leftarrow \quad \text{Cong}(A, B, P, Q) \quad \wedge \quad \text{Cong}(B, C, Q, R) \quad \wedge$
$\text{Eqangle}(B, A, B, C, Q, P, Q, R) \wedge \text{Ncoll}(A, B, C)$

37. $\text{Eqangle}(A, B, A, C, P, Q, P, R) \quad \leftarrow \quad \text{Cong}(A, B, P, Q) \wedge \text{Cong}(B, C, Q, R) \wedge$
$\text{Eqangle}(B, A, B, C, Q, P, Q, R) \wedge \text{Ncoll}(A, B, C)$

38. $\text{Eqangle}(C, A, C, B, R, P, R, Q) \quad \leftarrow \quad \text{Cong}(A, B, P, Q) \wedge \text{Cong}(B, C, Q, R) \wedge$
$\text{Eqangle}(B, A, B, C, Q, P, Q, R) \wedge \text{Ncoll}(A, B, C)$

39. $\text{Eqangle}(A, B, A, C, P, Q, P, R) \quad \leftarrow \quad \text{Eqangle}(B, A, B, C, Q, P, Q, R) \quad \wedge$
$\text{Eqangle}(C, A, C, B, R, P, R, Q) \wedge \text{Ncoll}(A, B, C)$

40. $\text{Eqratio}(A, B, P, Q, B, C, Q, R) \quad \leftarrow \quad \text{Eqangle}(B, A, B, C, Q, P, Q, R) \quad \wedge$
$\text{Eqangle}(C, A, C, B, R, P, R, Q) \wedge \text{Ncoll}(A, B, C)$

41. $\text{Eqratio}(B, C, Q, R, C, A, R, P) \quad \leftarrow \quad \text{Eqangle}(B, A, B, C, Q, P, Q, R) \quad \wedge$
$\text{Eqangle}(C, A, C, B, R, P, R, Q) \wedge \text{Ncoll}(A, B, C)$

42. $\text{Eqratio}(C, A, R, P, A, B, P, Q) \quad \leftarrow \quad \text{Eqangle}(B, A, B, C, Q, P, Q, R) \quad \wedge$
$\text{Eqangle}(C, A, C, B, R, P, R, Q) \wedge \text{Ncoll}(A, B, C)$

43. $\text{Cong}(B, C, Q, R) \quad \leftarrow \quad \text{Eqangle}(B, A, B, C, Q, P, Q, R) \quad \wedge$
$\text{Eqangle}(C, A, C, B, R, P, R, Q) \wedge \text{Ncoll}(A, B, C) \wedge \text{Cong}(A, B, P, Q)$

44. $\text{Cong}(A, C, P, R) \quad \leftarrow \quad \text{Eqangle}(B, A, B, C, Q, P, Q, R) \quad \wedge$
$\text{Eqangle}(C, A, C, B, R, P, R, Q) \wedge \text{Ncoll}(A, B, C) \wedge \text{Cong}(A, B, P, Q)$

45. $\text{Eqangle}(A, B, A, C, P, Q, P, R) \quad \leftarrow \quad \text{Eqangle}(B, A, B, C, Q, P, Q, R) \quad \wedge$
$\text{Eqangle}(C, A, C, B, R, P, R, Q) \wedge \text{Ncoll}(A, B, C) \wedge \text{Cong}(A, B, P, Q)$

46. $\text{Cong}(A, C, P, R) \quad \leftarrow \quad \text{Eqangle}(B, A, B, C, Q, R, Q, P) \quad \wedge$
$\text{Eqangle}(C, A, C, B, R, Q, R, P) \wedge \text{Ncoll}(A, B, C) \wedge \text{Cong}(A, B, P, Q)$

47. $\text{Cong}(B, C, Q, R) \quad \leftarrow \quad \text{Eqangle}(B, A, B, C, Q, R, Q, P) \quad \wedge$
$\text{Eqangle}(C, A, C, B, R, Q, R, P) \wedge \text{Ncoll}(A, B, C) \wedge \text{Cong}(A, B, P, Q)$

48. $\text{Eqangle}(A, B, A, C, P, Q, P, R) \quad \leftarrow \quad \text{Eqangle}(B, A, B, C, Q, R, Q, P) \quad \wedge$
$\text{Eqangle}(C, A, C, B, R, Q, R, P) \wedge \text{Ncoll}(A, B, C) \wedge \text{Cong}(A, B, P, Q)$

49. $\text{Eqangle}(A, B, A, C, P, Q, P, R) \quad \leftarrow \quad \text{Eqratio}(B, A, B, C, Q, P, Q, R) \quad \wedge$
$\text{Eqratio}(C, A, C, B, R, P, R, Q) \wedge \text{Ncoll}(A, B, C)$

50. $\text{Eqratio}(C, A, R, P, A, B, P, Q) \quad \leftarrow \quad \text{Eqratio}(B, A, B, C, Q, P, Q, R) \quad \wedge$
$\text{Eqratio}(C, A, C, B, R, P, R, Q) \wedge \text{Ncoll}(A, B, C)$

51. $\text{Eqangle}(B, A, B, C, Q, P, Q, R) \quad \leftarrow \quad \text{Eqratio}(B, A, B, C, Q, P, Q, R) \quad \wedge$
$\text{Eqratio}(C, A, C, B, R, P, R, Q) \wedge \text{Ncoll}(A, B, C)$

52. $\text{Eqangle}(C, A, C, B, R, P, R, Q) \quad \leftarrow \quad \text{Eqratio}(B, A, B, C, Q, P, Q, R) \quad \wedge$
$\text{Eqratio}(C, A, C, B, R, P, R, Q) \wedge \text{Ncoll}(A, B, C)$

53. $\text{Eqangle}(A, B, A, C, P, Q, P, R) \quad \leftarrow \quad \text{Eqratio}(B, A, B, C, Q, P, Q, R) \quad \wedge$
$\text{Eqangle}(B, A, B, C, Q, P, Q, R) \wedge \text{Ncoll}(A, B, C)$

54. $\text{Eqangle}(C, A, C, B, R, P, R, Q) \quad \leftarrow \quad \text{Eqratio}(B, A, B, C, Q, P, Q, R) \quad \wedge$
$\text{Eqangle}(B, A, B, C, Q, P, Q, R) \wedge \text{Ncoll}(A, B, C)$

55. $\text{Eqratio}(B, C, C, A, Q, R, R, P) \quad \leftarrow \quad \text{Eqratio}(B, A, B, C, Q, P, Q, R) \quad \wedge$
$\text{Eqangle}(B, A, B, C, Q, P, Q, R) \wedge \text{Ncoll}(A, B, C)$

56. $\text{Eqratio}(C, A, A, B, R, P, P, Q) \quad \leftarrow \quad \text{Eqratio}(B, A, B, C, Q, P, Q, R) \quad \wedge$
$\text{Eqangle}(B, A, B, C, Q, P, Q, R) \wedge \text{Ncoll}(A, B, C)$

57. $\text{Cong}(B, C, Q, R) \quad \leftarrow \quad \text{Eqratio}(B, A, B, C, Q, P, Q, R) \quad \wedge$
$\text{Eqratio}(C, A, C, B, R, P, R, Q) \wedge \text{Ncoll}(A, B, C) \wedge \text{Cong}(A, B, P, Q)$

58. $\text{Cong}(A, C, P, R) \quad \leftarrow \quad \text{Eqratio}(B, A, B, C, Q, P, Q, R) \quad \wedge$
$\text{Eqratio}(C, A, C, B, R, P, R, Q) \wedge \text{Ncoll}(A, B, C) \wedge \text{Cong}(A, B, P, Q)$

59. $\text{Eqangle}(A, B, A, C, P, Q, P, R) \quad \leftarrow \quad \text{Eqratio}(B, A, B, C, Q, P, Q, R) \quad \wedge$
$\text{Eqratio}(C, A, C, B, R, P, R, Q) \wedge \text{Ncoll}(A, B, C) \wedge \text{Cong}(A, B, P, Q)$

60. $\text{Eqangle}(B, A, B, C, Q, P, Q, R) \quad \leftarrow \quad \text{Eqratio}(B, A, B, C, Q, P, Q, R) \quad \wedge$
$\text{Eqratio}(C, A, C, B, R, P, R, Q) \wedge \text{Ncoll}(A, B, C) \wedge \text{Cong}(A, B, P, Q)$

61. $\text{Eqangle}(C, A, C, B, R, Q, R, P) \quad \leftarrow \quad \text{Eqratio}(B, A, B, C, Q, P, Q, R) \quad \wedge$
$\text{Eqratio}(C, A, C, B, R, P, R, Q) \wedge \text{Ncoll}(A, B, C) \wedge \text{Cong}(A, B, P, Q)$

62. $\text{Para}(M, N, A, B) \quad \leftarrow \quad \text{Para}(A, B, C, D) \wedge \text{Coll}(M, A, D) \wedge \text{Coll}(N, B, C) \wedge$
$\text{Eqratio}(M, A, M, D, N, B, N, C) \wedge \text{Sameside}(M, A, D, N, B, C)$

63. $\text{Eqratio}(M, A, M, D, N, B, N, C) \quad \leftarrow \quad \text{Para}(A, B, C, D) \wedge \text{Coll}(M, A, D) \wedge$
$\text{Coll}(N, B, C) \wedge \text{Para}(M, N, A, B)$

These rules are adapted from those in AlphaGeometry. [11]

---

[11] https://github.com/google-deepmind/alphageometry/blob/main/jgex_ag_231.txt

# B    DATA EXAMPLES FROM GEOILP

## B.1    PREDICATE INVENTION

The example below requires the invention of Eqangle

Background Knowledge:

- Cong(D8,A8,D8,B8)
- Cong(D8,B8,D8,C8)
- Cong(D8,E8,D8,A8)
- Coll(C8,A8,F8)
- Perp(E8,F8,A8,C8)
- Coll(G8,A8,B8)
- Perp(E8,G8,A8,B8)
- Ncoll(F8,G8,A8)
- Cong(D9,A9,D9,B9)
- Cong(D9,B9,D9,C9)
- Cong(D9,E9,D9,A9)
- Coll(C9,A9,F9)
- Perp(E9,F9,A9,C9)
- Coll(G9,A9,B9)
- Perp(E9,G9,A9,B9)
- Ncoll(F9,G9,A9)

Positive Examples:

- Cyclic(C8,E8,A8,B8)
- Cyclic(E8,A8,F8,G8)
- Cyclic(C9,E9,A9,B9)
- Cyclic(E9,A9,F9,G9)

Proof steps:

- Cyclic(C8,E8,A8,B8)  ←  Cong(D8,A8,D8,B8)  ∧  Cong(D8,E8,D8,A8)  ∧ Cong(D8,B8,D8,C8)
- Coll(F8,A8,C8) ← Coll(C8,A8,F8)
- Coll(G8,A8,B8) ← Coll(G8,A8,B8)
- Eqangle(A8,C8,E8,F8,A8,B8,E8,G8) ← Perp(E8,G8,A8,B8) ∧ Perp(E8,F8,A8,C8)
- Eqangle(F8,A8,F8,E8,G8,A8,G8,E8)  ←  Coll(F8,A8,C8) ∧ Coll(G8,A8,B8) ∧ Eqangle(A8,C8,E8,F8,A8,B8,E8,G8)
- Cyclic(E8,A8,F8,G8) ← Eqangle(F8,A8,F8,E8,G8,A8,G8,E8) ∧ Ncoll(F8,G8,A8)
- Cyclic(C9,E9,A9,B9)  ←  Cong(D9,A9,D9,B9)  ∧  Cong(D9,E9,D9,A9)  ∧ Cong(D9,B9,D9,C9)
- Coll(F9,A9,C9) ← Coll(C9,A9,F9)
- Coll(G9,A9,B9) ← Coll(G9,A9,B9)
- Eqangle(A9,C9,E9,F9,A9,B9,E9,G9) ← Perp(E9,G9,A9,B9) ∧ Perp(E9,F9,A9,C9)
- Eqangle(F9,A9,F9,E9,G9,A9,G9,E9)  ←  Coll(F9,A9,C9) ∧ Coll(G9,A9,B9) ∧ Eqangle(A9,C9,E9,F9,A9,B9,E9,G9)
- Cyclic(E9,A9,F9,G9) ← Eqangle(F9,A9,F9,E9,G9,A9,G9,E9) ∧ Ncoll(F9,G9,A9)

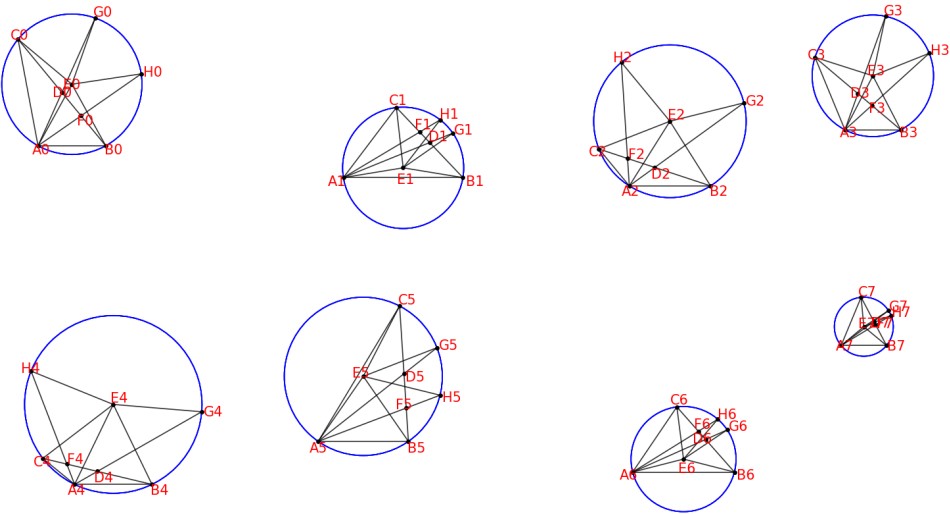

Figure 2: An example of image-form background knowledge.

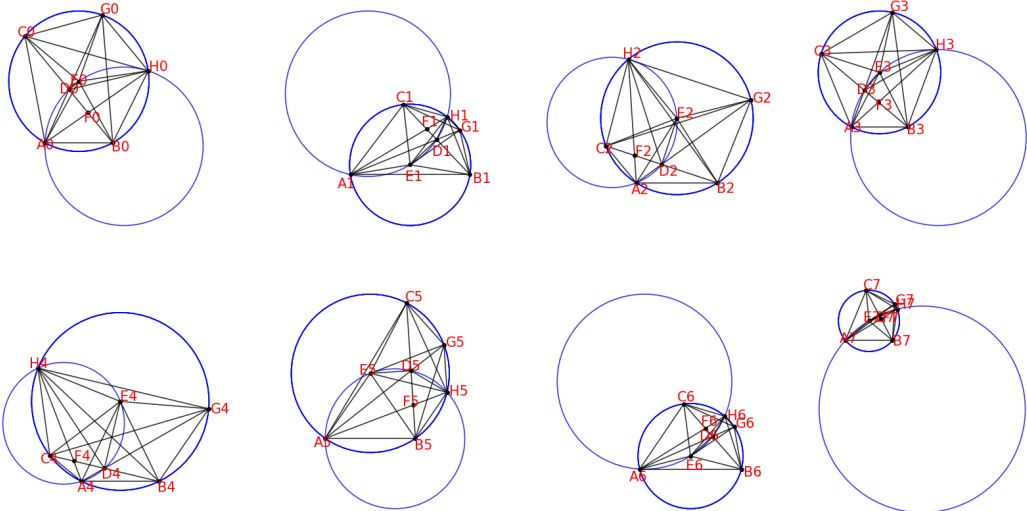

Figure 3: An example of image-form background knowledge + positive examples.

## B.2 RAW SENSORY DATA

See Figure 2 for an example of image-form background knowledge. See Figure 3 for an example of image-form conclusions, which is a combination of background knowledge and positive examples.

## C TRANSLATING GEOILP INTO NATURAL-LANGUAGE FORM

For readers interested in the induction ability of large language models (LLMs), we provide guidance for translating GeoILP into natural-language form.

The prompt fed to LLMs should consist of three parts: task description, task data, and a command requiring LLMs to induce a hypothesis. The difficult part is the task data, containing background knowledge and positive & negative examples, which are ground atoms. The translation from symbolic atoms to natural-language forms varies in different domains. For *plane geometry*, AlphaGeom-

etry (Trinh et al., 2024) provides several templates. For example, $\mathrm{Coll}(A, B, C)$ is translated into *A,B,C are collinear*. All the predicates used in GeoILP can be found in AlphaGeometry, and thus all GeoILP's atoms can be translated into natural-language forms. In addition, to enforce LLMs generating Horn rules (in natural-language forms), the command could ask LLMs to generate rules in the form of "If ... and ..., then ...", where each "..." corresponds to a ground atom. "..." can be repeated multiple times and must be concatenated by "and".

