# OpenReview forum: "GeoILP: A Synthetic Dataset to Guide Large-Scale Rule Induction"
_ICLR.cc/2025/Conference — ICLR 2025 Poster_

### Official Review · Reviewer_swwC · 2024-10-28

**Soundness:** 2
**Presentation:** 2
**Contribution:** 2
**Rating:** 3
**Confidence:** 2

**Summary:**

This paper proposed a new dataset in the geometry domain to help the researcher enhance the inductive logic programming (ILP) models. The authors indicated that there is no reference hypothesis in the current ILP datasets. This paper proposes a larger-scale ILP dataset with reference hypotheses. This paper also describes the algorithm for how to generate the synthesis data and the corresponding reference.

However, the paper is borderline rejected for the following reasons: (1) The motivation of the paper is not explicitly discussed in the paper. (2) The Experiments are not explicitly investigated in the paper. Hence, the overall contribution of the paper is limited. (3) Some terminology is vague to use such as learning from raw data and open-world assumption.

**Strengths:**

The paper builds a novel ILP dataset to boost the development of the ILP community development. The proposed ILP is very challenging based on the statements of the paper. In addition, the authors present the methodology for generating the proposed ILP dataset.

**Weaknesses:**

1. Based on the structure of the paper, only Section 5 describes the proposed GeoILP. The rest of the Sections look like a survey to describe the development of the ILP methodology. Hence, the contribution of the paper including the method to generate the datasets and the evaluation of the proposed dataset for proving some properties such as learning recursive rules and long variables rules with the existing ILP models is still limited.
2. When learning from raw data, the authors only discussed learning rules with the help of a pre-trained perception model. However, some discussions about learning from raw data directly without the symbolization process by the perception model are missing.

**Questions:**

1. Why is this dataset helpful in solving the ILP problem?
    1. Having reference hypotheses to guide the evaluation of the ILP model is not essential in all senses. In some ILP datasets proposed by [1] or FB15KSelected, the knowledge graph is easy to understand. Hence, there is no need to have a reference in addition.
    2. Besides, some ILP models support precision and recall as quantitative metrics to evaluate the learned rules from data [3].
    3. Furthermore, when generating a set of rules, how to evaluate the performance of an ILP model based on the proposed reference hypotheses. Is the reference complete based on the background knowledge, positive examples, and negative examples?
2. Based on these proposed datasets, the authors mentioned that no one ILP model can successfully learn rules from the GeoILP. The results are further explained in Section 6.2. The symbolic ILP models can not learn even one rule in the *basic level* setting. However, there is no explicit explanation about the basic level in line 520 page 10.  In addition, in line 524 on page 10, the authors stated that three neuro-symbolic models cannot solve the GeoILP because of the features of these ILP models. However, some neuro-symbolic ILP models can learn rules with three or more body atoms and any arities of a predicate [2]. The authors should also analyze more ILP models in Experiments to investigate the current performance of the ILP models.
3. In addition, there is no reference in the Open-world assumption paragraph of Section  4.2. In addition, the open-world assumption is not clearly explained as to why the open-world assumption is related to intentional and extensional predicates in line 428 on page 8. The knowledge base uses the open-world assumption to determine the Boolean value of a ground atom, which is defined in line 257 on page 5. However, in line 428, the open-world assumption is applied based on the rule format. Hence, the authors should explain more about the connections between open-world assumptions and the format of rules.

Reference:
[1] Richard Evans, Edward Grefenstette: Learning Explanatory Rules from Noisy Data. J. Artif. Intell. Res. 61: 1-64 (2018)
[2] Xujie Si, Mukund Raghothaman, Kihong Heo, Mayur Naik: Synthesizing Datalog Programs using Numerical Relaxation. IJCAI 2019: 6117-6124
[3] Tim Rocktäschel, Sebastian Riedel: End-to-end Differentiable Proving. NIPS 2017: 3788-3800

---

> ### Author Response · Authors · 2024-11-21
>
> We appreciate your insightful comments and constructive suggestions, which greatly contribute to improving our paper.
>
> Below, we respond to your questions point by point and detail how we revised the paper based on your suggestions.
> **(The cited references or texts below correspond to the revised paper, which has already been uploaded to the review system.)**
>
> > **Question 1.1**: Having reference hypotheses to guide the evaluation of the ILP model is not essential in all senses. In some ILP datasets proposed by [1] or FB15KSelected, the knowledge graph is easy to understand. Hence, there is no need to have a reference in addition.
>
> - **Having reference hypotheses is not to guide the evaluation but rather to guide the development of ILP methods.** Although several knowledge graphs (i.e., *facts*) are easy to understand, the *rules* generating them remain unclear. Therefore, researchers would wonder where to improve if an ILP method fails on these knowledge graphs.
>   In contrast, having reference hypotheses will tell researchers that the current ILP method cannot handle, for example, higher predicate arity, recursion, or predicate constraints.
>   Such spirit has been discussed in the 4th paragraph of *Introduction* and in *Section 2.2*.
>
> - We add the citation of FB15K (Bordes et al., 2013) and FB15KSelected (Toutanova & Chen, 2015) in *Real-world datasets, Section 2.2* ([1] has been already cited).
>
>
> > **Question 1.2**: Besides, some ILP models support precision and recall as quantitative metrics to evaluate the learned rules from data [3].
>
> - All evaluation methods designed for ILP are accepted for GeoILP, including the one you mentioned.
>   As explained above, reference hypotheses primarily contribute to guiding the development of ILP methods instead of evaluation.
>
> - We add discussion and citation of [3] (Rocktaschel & Riedel, 2017) in *Neural-symbolic methods, Section 2.1* and *Neuro-symbolic, Section 6.2*.
>
>
> > **Question 1.3**: Furthermore, when generating a set of rules, how to evaluate the performance of an ILP model based on the proposed reference hypotheses. Is the reference complete based on the background knowledge, positive examples, and negative examples?
>
> - As explained above, the reference hypotheses in GeoILP do not contribute to the evaluation.
>   In addition, comparing the reference hypotheses and the learned hypotheses would be an interesting analysis, revealing the difference in induction preference between human experts and machine learning models.
>
> - The reference hypotheses are *incomplete* (but *sound*) for positive examples since they are *incomplete* for the entire *plane geometry*. The reference hypotheses are *complete* (but not *sound*) for negative examples for the same reason.
>
>   - Note that determining a complete set of rules is out of the scope of our work, and our rule set follows AlphaGeometry (Trinh et al., 2024), which achieves SOTA performance on the latest IMO geometry problems.
>
>   - This is another source of noise. In the revised paper, we emphasize it in *Noisy data, Section 5.3*.
>
>
> > **Question 2**: Based on these proposed datasets, the authors mentioned that no one ILP model can successfully learn rules from the GeoILP. The results are further explained in Section 6.2. The symbolic ILP models can not learn even one rule in the *basic level* setting. However, there is no explicit explanation about the basic level in line 520 page 10. In addition, in line 524 on page 10, the authors stated that three neuro-symbolic models cannot solve the GeoILP because of the features of these ILP models. However, some neuro-symbolic ILP models can learn rules with three or more body atoms and any arities of a predicate [2]. The authors should also analyze more ILP models in Experiments to investigate the current performance of the ILP models.
>
> - The *basic level* has been described in *Section 6.1* and *Table 3*.
>
> - Per your suggestion, we test Difflog [2] on the *basic* level of GeoILP. We use the implementation and recommended parameter setting [here](https://github.com/petablox/difflog) and conduct experiments on a server with at least 500GB of memory.
>   However, it throws *out of memory error* in just a few minutes. Note that our memory usage is an order of magnitude larger than in the original paper (64GB). Therefore, we argue that Difflog should be significantly enhanced before being applicable to GeoILP. We add discussion and citation in *Neuro-symbolic, Section 6.2* and *Neural-symbolic methods, Section 2.1*.

---

> ### Author Response · Authors · 2024-11-21
>
> > **Question 3**: In addition, there is no reference in the Open-world assumption paragraph of Section 4.2. In addition, the open-world assumption is not clearly explained as to why the open-world assumption is related to intentional and extensional predicates in line 428 on page 8. The knowledge base uses the open-world assumption to determine the Boolean value of a ground atom, which is defined in line 257 on page 5. However, in line 428, the open-world assumption is applied based on the rule format. Hence, the authors should explain more about the connections between open-world assumptions and the format of rules.
>
> - We add a reference (Reiter, 1981) in *Open-world assumption of Section 4.2*, which discusses closed-world assumption and open-world assumption.
>
> - To be clear, we rephrase *Noisy data, Section 5.3* in the revised paper. Here is an explanation of why GeoILP assumes OWA:
>
>   - When an atom appears in BK, it is assumed to be true (and its predicate is called *extensional*).
>
>   - In CWA, the atoms not given in BK are assumed to be false. However, in GeoILP, several true atoms are not directly given in BK, but deduced by forward-chaining instead. This exactly corresponds to OWA.
>
>   - When an atom is deduced by forward-chaining, a ground rule with this atom as the head must exist. Therefore, the predicate of this atom is called *intensional*. This is the reason why OWA is related to intensional and extensional predicates.
>
>
>   The definitions of *extensional* and *intensional* at the end of *Section 3.2* are no longer needed as we remove *extensional* and *intensional* from *Noisy data, Section 5.3*. Therefore, we also remove these definitions in the revised paper.
>
> - Also in *Noisy data, Section 5.3*, we change the second source of noisy data. In the original paper, we manually added noise to the data, which we found not necessary since two sources of noise already exist. The first source is OWA, and the second source is the incomplete rule set, as explained in our response to your *Question 1.3*.
>
>
> > **Weaknesses 1**: Based on the structure of the paper, only Section 5 describes the proposed GeoILP. The rest of the Sections look like a survey to describe the development of the ILP methodology. Hence, the contribution of the paper including the method to generate the datasets and the evaluation of the proposed dataset for proving some properties such as learning recursive rules and long variables rules with the existing ILP models is still limited.
>
> - Regarding "The rest of the Sections look like a survey to describe the development of the ILP methodology", we argue it is *Section 4* that makes you think so. If so, here are several reasons why *Section 4* is crucial.
>
>   - It provides further background definitions related to ILP (such as the definition of recursion, OWA, etc.), which are needed in the following sections.
>
>   - It explains why those language biases are essential in real-world applications. Without these explanations, readers would wonder if these language biases are only meaningful for GeoILP and make no sense for broader applications.
>
>   - It provides reasons why we need to cover these language biases in GeoILP, i.e., current ILP methods do not support them well, and we need a testbed for them.
>
>
>   We can merge this section with *Section 5* but we argue that writing in the current way is clearer and friendly for readers unfamiliar with ILP.
>
> - Regarding the method to generate the datasets, we are willing to provide more details if you could provide more detailed guidance.
>
> - As described above, we have added experiments following your suggestions. We are willing to add more experiments if other ILP methods applicable to GeoILP exist.
>
>
> > **Weaknesses 2**: When learning from raw data, the authors only discussed learning rules with the help of a pre-trained perception model. However, some discussions about learning from raw data directly without the symbolization process by the perception model are missing.
>
> - One of our paper's and ILP research's goals is to build explainable AI, as described in *Introduction*. We are unaware of any method that can both produce explainable models and "learning from raw data directly without the symbolization process by the perception model".
>
> - According to the suggestions from *Reviewer Y6Wf*, we add several citations (Evans (2020); Evans et al. (2021; 2022)) at the end of *Section 4.4 UNABLE TO LEARN FROM RAW INPUT*. However, these methods still require a symbolization process.

---

> ### Comment · Reviewer_swwC · 2024-11-23
>
> Thank you for your response. I find the hypothesis quite complex. In the ILP domain, using terms in atoms could also be a potential solution. However, the discussion regarding using terms to address GeoILP is missing. I wonder if leveraging terms alongside existing ILP methods might solve some of the challenges in GeoILP.
>
> Additionally, the authors mentioned using Popper (Cropper & Morel, 2021a) to verify its ability to produce correct results, citing its support for predicate invention (Cropper & Morel, 2021b) and noise handling (Hocquette et al., 2024) in line 507. However, Section 6.1.1 of Cropper & Morel (2021a) indicates that Popper does not support predicate invention or noise handling as claimed. This raises concerns about whether the experiments were conducted accurately.
>
> In conclusion, I still believe that using the existing ILP model to solve highly complex ILP tasks, particularly those involving results with 8-arity predicates, is unfair. I think the limitation might not stem from ILP itself but rather from the method used to process the data, such as simplifying it through domain knowledge or transformations. I would like to maintain my current score.
>
> Reference:
>
> Andrew Cropper and Rolf Morel. Learning programs by learning from failures. Machine Learning, 110(4):801–856, 2021a.
>
> Andrew Cropper and Rolf Morel. Predicate invention by learning from failures. arXiv preprint arXiv:2104.14426, 2021b.
>
> Céline Hocquette, Andreas Niskanen, Matti J¨ arvisalo, and Andrew Cropper. Learning mdl logic programs from noisy data. In Proceedings of the AAAI Conference on Artificial Intelligence, volume 38, pp. 10553–10561, 2024.

---

> > ### Author Response · Authors · 2024-11-24
> >
> > We appreciate your active reply. Here are our responses to your further concerns.
> >
> > > I find the hypothesis quite complex. In the ILP domain, using terms in atoms could also be a potential solution. However, the discussion regarding using terms to address GeoILP is missing. I wonder if leveraging terms alongside existing ILP methods might solve some of the challenges in GeoILP.
> >
> > - *Terms* include *constants*, *variables*, and *functions* of them. Since *constants* and *variables* already appear in GeoILP's hypotheses, we think that your question is about "how to use *functions* to address GeoILP".
> > - In Inductive Logic Programming, Logic Programming, and many logic-related research domains, regarding a function in the form $B=f(A)$ as a relation $P(A,B)$, which require the uniqueness of the second argument, is common. In other words, all functions can be transformed into relations.
> > - We have already mentioned *functions* in *Predicate constraint, 4.1 HAND-CRAFTED LANGUAGE BIAS* and *Predicate, 5.3 DATASET FEATURES*. In *Predicate constraint, 4.1 HAND-CRAFTED LANGUAGE BIAS*, we have cited Cropper & Hocquette
> >   (2023), which handled *functions*. In the Table 1 of Cropper & Hocquette
> >   (2023), they also regard *functions* as *relations* and use Horn goals to constrain *relations* to *functions*.
> >
> > > Additionally, the authors mentioned using Popper (Cropper & Morel, 2021a) to verify its ability to produce correct results, citing its support for predicate invention (Cropper & Morel, 2021b) and noise handling (Hocquette et al., 2024) in line 507. However, Section 6.1.1 of Cropper & Morel (2021a) indicates that Popper does not support predicate invention or noise handling as claimed. This raises concerns about whether the experiments were conducted accurately.
> >
> > - Popper initially did not support predicate invention and noise handling (Cropper & Morel, 2021a), but has been extended to support them in the following papers (Cropper & Morel, 2021b; Hocquette et al., 2024).
> >
> > - This is clearly stated in Popper's [code repository](https://github.com/logic-and-learning-lab/Popper/tree/v4.3.0). See *Noisy examples* and *Predicate invention* sections in the READMD.md of the code repository.
> >
> >   - This link has been cited in the footnote 7 in our paper.
> >
> >   - Note that, in the code repository, *noisy examples* section cited the preprint version of Hocquette et al. (2024). In our paper, we cited its version published in AAAI.
> >
> >
> > > In conclusion, I still believe that using the existing ILP model to solve highly complex ILP tasks, particularly those involving results with 8-arity predicates, is unfair. I think the limitation might not stem from ILP itself but rather from the method used to process the data, such as simplifying it through domain knowledge or transformations. I would like to maintain my current score.
> >
> > - Although using domain knowledge might be a possible way for GeoILP, handling various language biases with tremendous hand-crafted expert design does not accord with the modern spirit of machine learning, which significantly impedes the development of ILP towards broader applications.
> >
> >   - As discussed in the 2nd paragraph of *Introduction*, automatic learning without expert knowledge is one of the primary reasons why modern machine learning (e.g., deep learning) gains success. However, as described throughout our paper, many current methods require hand-crafted rule templates to define language bias task by task (and domain by domain). Even if several methods support relatively general templates (Glanois et al., 2022), they are limited to restricted language biases, such as low-arity predicates, not supporting mutual recursion.
> >
> >   - We also express the prospect of developing and evaluating ILP methods in an expert-free paradigm, i.e., no heavily hand-crafted design, just like deep learning. This has been described at the end of the 4th paragraph of *Introduction* and throughout the paper.
> >
> >   - In summary, GeoILP challenges existing hand-crafted ILP methods. We argue that, just as hand-crafted feature engineering is rarely used nowadays in large-scale AI applications, hand-crafted ILP will be out-of-date one day, and automatic ILP will be one of the appropriate ways toward large-scale explainable AI.
> >
> >   - Besides, whatever the way to deal with large-scale complex ILP problems, it is out of the scope of a dataset paper. GeoILP is a good testbed for large-scale complex ILP since it covers all the challenging language biases and is large enough.
> >
> >
> > We are more than willing to provide further information if our responses do not resolve your concerns.

---

### Official Review · Reviewer_Prcn · 2024-11-02

**Soundness:** 3
**Presentation:** 3
**Contribution:** 4
**Rating:** 8
**Confidence:** 4

**Summary:**

This paper presents a novel Inductive Logic Programming (ILP) dataset to evaluate ILP algorithms and systems. The presented GeoILP tasks try to formulate geometrical background knowledge as logic programs, and the motivation is to learn geometrical theories/concepts from them. Moreover, the authors also visualised all the tasks and produced images for the problems. The difficulty of the tasks is varied, ranging from simple questions to IMO-level questions.

**Strengths:**

- The paper discussed the challenges for the current ILP area in detail, and the authors clearly understood the intrinsic disadvantages of ILP, thus the dataset is designed to motivate the community to resolve these long-neglected issues.
- The authors have covered extensive related works, and the paper is well structured and well written.

**Weaknesses:**

- The design of the dataset is thoughtful, however, it is still like the previous ILP tasks, which are not very accessible to the ICLR community. For example, the representation of Logic Programming or Prolog is not user-friendly for normal users. The definition and theorems in plane geometry described with logic programs in this paper sometimes are difficult to understand.
- Prolog is not a popular language for theorem proving, modern machine learning techniques such as LLMs could not handle them well enough.
- The experiments in this paper are not enough, it is only experimented with Popper.

**Questions:**

- Induction will be a very challenging problem for LLMs, would it be possible to extend the dataset and make a natural language version, so that people can compare ILP with LLMs?
- Why not use formal languages that are designed specifically for automated proof, such as lean4 or Coq?

---

> ### Author Response · Authors · 2024-11-21
>
> We appreciate your insightful comments and constructive suggestions, which greatly contribute to improving our paper.
>
> Below, we respond to your questions point by point and detail how we revised the paper based on your suggestions.
> **(The cited references or texts below correspond to the revised paper, which has already been uploaded to the review system.)**
>
> > **Question 1**: Induction will be a very challenging problem for LLMs, would it be possible to extend the dataset and make a natural language version, so that people can compare ILP with LLMs?
>
> Translating GeoILP tasks into natural language form is possible with a template-based text-generating technique. In the revised paper, we add guidance in Appendix C and cite it at the end of *Section 5.4 CONSTRUCTING RAW INPUT*.
>
> > **Question 2**: Why not use formal languages that are designed specifically for automated proof, such as lean4 or Coq?
> >
> > **Weaknesses 2**: Prolog is not a popular language for theorem proving, modern machine learning techniques such as LLMs could not handle them well enough.
>
> We use Prolog as the formal language for the following two reasons:
>
> 1. Prolog is used in many geometry deduction systems, such as GEX (Chou et al., 2000), JGEX (Ye et al., 2010a;b; 2011), AlphaGeometry (Trinh et al., 2024). Among them, AlphaGeometry achieves SOTA performance on the latest IMO geometry problems and has been published in *Nature* in 2024. Note that AlphaGeometry also uses LLM to help with Prolog deduction. We follow these works to use Prolog, and determining the formal language to use is out of the scope of our paper.
>
> 2. A large majority of current ILP methods focuses on Horn clauses (e.g., Popper (Cropper & Morel, 2021a), HRI (Glanois et al., 2022)). We leave further studies on more expressive formal languages to future work.
>
>
> > **Weaknesses 1**: The design of the dataset is thoughtful, however, it is still like the previous ILP tasks, which are not very accessible to the ICLR community. For example, the representation of Logic Programming or Prolog is not user-friendly for normal users. The definition and theorems in plane geometry described with logic programs in this paper sometimes are difficult to understand.
>
> We acknowledge the fact that formal languages, such as Prolog, Coq, lean4, are not user-friendly. However, formal languages are still generally accepted in Math reasoning because of the possibility and simplicity (relative to natural languages) of strictly discussing their *soundness* and *completeness*.
>
> In addition, as listed in the Call for Papers of ICLR 2025, *neuro-symbolic & hybrid AI systems (physics-informed, logic & formal reasoning, etc.)* is an accepted topic. Therefore, our work should be accessible to a part of ICLR community.
>
> > **Weaknesses 3**: The experiments in this paper are not enough, it is only experimented with Popper.
>
> According to the suggestion from Reviewer *swwC*, we add experiments on Difflog (Si et al., 2019), a neural-symbolic model that supports arbitrary hypothesis space. We use the implementation and recommended parameter setting [here](https://github.com/petablox/difflog) and conduct experiments on a server with at least 500GB of memory.
> However, it throws *out of memory error* in just a few minutes. Note that our memory usage is an order of magnitude larger than in the original paper (64GB). Therefore, we argue that Difflog should be significantly enhanced before being applicable to GeoILP. We add discussion and citation in *Neuro-symbolic, Section 6.2* and *Neural-symbolic methods, Section 2.1*.
>
> The experimental section reveals that all current methods fail on GeoILP, which justifies the significance of GeoILP. We reserve the possibility that existing methods might solve GeoILP after considerable enhancement and extension (e.g., dramatically accelerating the training time of Popper or dramatically reducing the memory usage of Difflog), but this requires substantial research and thus is left to future work.

---

> > ### Comment · Reviewer_Prcn · 2024-11-24
> >
> > Thank you for the answers, I appreciate the revisions you made, especially the accessibility improvement for language models.

---

### Official Review · Reviewer_Qcx3 · 2024-11-03

**Soundness:** 2
**Presentation:** 3
**Contribution:** 3
**Rating:** 5
**Confidence:** 3

**Summary:**

The paper  introduce GeoILP, a large-scale synthetic dataset of diverse ILP tasks involving numerous aspects of language bias.
The dataset consists of geometry problems modeled from levels as varied as textbook exercises, promoting the development of methods that can handle complex language biases, higher arity, and multi-task learning.

**Strengths:**

1. The paper is clearly written.

2. The dataset GEOILP is  large-scale and mimics real-world data better than traditional closed-world datasets.

**Weaknesses:**

The article provides a lot of background information, which results in insufficient coverage of its own work.
Firstly, the experimental section is difficult to support the contributions of this paper.
Also, although Section 5 offers a detailed introduction to the content of its dataset, it does not effectively highlight the characteristics of its dataset in comparison to other datasets.

**Questions:**

How strong is the generalization capability of the neural-symbolic model trained on this dataset? Can it solve tasks beyond geometric problems?

**Details Of Ethics Concerns:**

None.

---

> ### Author Response · Authors · 2024-11-21
>
> We appreciate your insightful comments and constructive suggestions, which greatly contribute to improving our paper.
>
> Below, we respond to your questions point by point and detail how we revised the paper based on your suggestions.
> **(The cited references or texts below correspond to the revised paper, which has already been uploaded to the review system.)**
>
> > **Question**: How strong is the generalization capability of the neural-symbolic model trained on this dataset? Can it solve tasks beyond geometric problems?
>
> Any ILP model that performs well on GeoILP is presumed to perform well on solving tasks beyond geometric problems after training on the corresponding domains. Let's explain step by step:
>
> 1. Each ILP task consists of background knowledge and target examples, which may differ in different domains. We have acknowledged the **domain bias** of GeoILP in the *Discussion and Conclusion* section.
>
> 2. However, GeoILP covers all the challenging **language biases** in the ILP problem (as detailed in the *Limitations of Current ILP* section).
>
> 3. One of the crucial issues impeding broader applications of ILP is that, especially for neuro-symbolic ILP methods, several crucial language biases are not supported. This makes those methods **inapplicable** to many problems (e.g., most current neuro-symbolic methods do not support predicate arity greater than two).
>
> 4. Nevertheless, no dataset with various language biases exists. This is one of the most essential reasons why GeoILP is useful.
>
> 5. Therefore, any ILP model that performs well on GeoILP would have extraordinary abilities to deal with such various language biases. When training such models on other tasks/domains, it makes sense to believe in the effectiveness of such models due to their ability to handle various language biases.
>
> 6. Of course, the performance of these models may not be good enough on other tasks/domains because of the domain bias of GeoILP, i.e., the hypothesis space changes. We leave this to future work as the current crucial issue is those language biases.
>
>
> > **Weakness**: The article provides a lot of background information, which results in insufficient coverage of its own work. Firstly, the experimental section is difficult to support the contributions of this paper. Also, although Section 5 offers a detailed introduction to the content of its dataset, it does not effectively highlight the characteristics of its dataset in comparison to other datasets.
>
> 1. All the background information, including Section 3&4, is indispensable for the following reasons:
>
>   - It provides further background definitions related to ILP (such as the definition of recursion, OWA, etc.), which are needed in the following sections.
>
>   - It explains why those language biases are essential in real-world applications. Without these explanations, readers would wonder if these language biases are only meaningful for GeoILP and make no sense for broader applications.
>
>   - It provides reasons why we need to cover these language biases in GeoILP, i.e., current ILP methods do not support them well, and we need a testbed for them.
>
> 2. The experimental section reveals that all current methods fail on GeoILP, which justifies the significance of GeoILP. We reserve the possibility that existing methods might solve GeoILP after considerable enhancement and extension (e.g., dramatically accelerating the training time of Popper), but this requires substantial research and thus is left to future work.
>
> 3. According to the suggestion from Reviewer *swwC*, we add experiments on Difflog (Si et al., 2019), a neural-symbolic model that supports arbitrary hypothesis space. We use the implementation and recommended parameter setting [here](https://github.com/petablox/difflog) and conduct experiments on a server with at least 500GB of memory.
>   However, it throws *out of memory error* in just a few minutes. Note that our memory usage is an order of magnitude larger than in the original paper (64GB). Therefore, we argue that Difflog should be significantly enhanced before being applicable to GeoILP. We add discussion and citation in *Neuro-symbolic, Section 6.2* and *Neural-symbolic methods, Section 2.1*.
>
> 4. We have compared GeoILP with other datasets in Section 5.3.1 and Table 2. This table shows that GeoILP tremendously improves the complexity of ILP datasets.

---

### Official Review · Reviewer_Y6Wf · 2024-11-07

**Soundness:** 4
**Presentation:** 4
**Contribution:** 3
**Rating:** 8
**Confidence:** 3

**Summary:**

This paper introduces a large-scale synthetic dataset for inductive logic programming involving a number of challenging predicate language constructions.  The subject domain of geometry provides a rich set of predicates with symmetries, recursion, and constraints. Furthermore, when higher dimensional connective objects (e.g. line segments) are excluded from the background knowledge, predicates become black-box functions of the remaining universe of objects (e.g. points).  This setup provides an instructive setting in which to illustrate and investigate many prominent and unsolved challenges in ILP.

**Strengths:**

- This paper provides a reference dataset for investigating a number of important challenges in ILP for which no current datasets exist.
- The problem domain is intuitive, providing the potential to be expanded by others as additional points of interest arise.
- Open challenges in ILP are well articulated, and the role of this dataset in providing a means to evaluate future developments in the field is conveyed clearly.

**Weaknesses:**

- The predicate arity section may not be ideally motivated.  For example, the (teacher, subject, student) relationship would probably be addressed in practice by making the subject a predicate, since a relatively small set of subjects at that level of granularity exist.  While the authors' observation is valid, it but might not be compelling to someone unfamiliar with the nuance.
- In addition to combinatorial techniques referenced heavily in the writeup, there are a wide variety of published neuro symbolic techniques.  One of the few highlighted in this paper is by Evans and Grefenstette.  Evans et. all discuss raw data challenges [1], and his dissertation [2] has extensive discussion of challenges and approach.

[1] Proceedings of the Thirty-First International Joint Conference on Artificial Intelligence (IJCAI-22)
[2] Kant's Cognitive Architecture

**Questions:**

The statement "Enabling recursion is expensive for symbolic ILP, while neuro-symbolic ILP does not support mutual recursion" cites a singe neuro-symbolic method that presumably does not support mutual recursion.  Is there a theorem that says that no neuro-symbolic method can support mutual recursion?

Is there a reason to exclude the wider neuro-symbolic techniques that have been explored?

Can you discuss the limitations of binary predicates in more detail?  I think this deserves more attention.

---

> ### Author Response · Authors · 2024-11-21
>
> We appreciate your insightful comments and constructive suggestions, which greatly contribute to improving our paper.
>
> Below, we respond to your questions point by point and detail how we revised the paper based on your suggestions.
> **(The cited references or texts below correspond to the revised paper, which has already been uploaded to the review system.)**
>
> > **Question 1**: The statement "Enabling recursion is expensive for symbolic ILP, while neuro-symbolic ILP does not support mutual recursion" cites a singe neuro-symbolic method that presumably does not support mutual recursion. Is there a theorem that says that no neuro-symbolic method can support mutual recursion?
> >
> > Is there a reason to exclude the wider neuro-symbolic techniques that have been explored?
>
> There is no theorem saying that no neuro-symbolic method can support mutual recursion, but most neuro-symbolic methods indeed do not support mutual recursion due to their model structure. We only cite one method as it is the SOTA, not only in the sense of the highest experimental performance but also in the sense of its advantage of no requirement for defining language bias task by task. Note that the latter is one of the crucial points for broader applications of ILP.
>
> However, we found this statement to be unfair as there are neuro-symbolic ILP methods that can support mutual recursion (Campero et al., 2018, Si et al.,
> 2019). To be more rigorous, we change the statement from "neuro-symbolic ILP does not support mutual recursion" to "the state-of-the-art neuro-symbolic ILP method does not support mutual recursion".
>
> > **Weakness 1**: The predicate arity section may not be ideally motivated. For example, the (teacher, subject, student) relationship would probably be addressed in practice by making the subject a predicate, since a relatively small set of subjects at that level of granularity exist. While the authors' observation is valid, it but might not be compelling to someone unfamiliar with the nuance.
> >
> > **Question 2**: Can you discuss the limitations of binary predicates in more detail? I think this deserves more attention.
>
> Degrading the ternary predicate, say $Course(teacher, subject, student)$, to binary predicates, say $Math(teacher, student)$, $Physics(teacher, student)$, $Chemistry(teacher, student)$, ......, at least have following negative impacts:
>
> - Increase the number of predicates. This might be difficult for some sorts of ILP methods as the hypothesis space size would increase combinatorially.
>
> - Turn *subject* objects into predicates. In practice, the predicate set and the universe of objects are greatly determined according to task settings. Here is a case where treating *subjects* as objects is more natural than treating them as predicates: a task additionally involving another relation asserting that a *student* is good at a *subject*, say $GoodAt(student, subject)$. The most crucial benefit of treating *subjects* as objects is the ability to establish many other relations between *subjects* and many other types of objects.
>
>
> To reduce such confusion, we add $GoodAt(student, subject)$ along with $Course(teacher, subject, student)$ in that paragraph.
>
> > **Weakness 2**: In addition to combinatorial techniques referenced heavily in the writeup, there are a wide variety of published neuro symbolic techniques. One of the few highlighted in this paper is by Evans and Grefenstette. Evans et. all discuss raw data challenges [1], and his dissertation [2] has extensive discussion of challenges and approach.
>
> Thanks for pointing out these references. In the revised paper, we add these citations at the end of *Section 4.4 UNABLE TO LEARN FROM RAW INPUT*.

---

> > ### Comment · Reviewer_Y6Wf · 2024-11-22
> >
> > I appreciate the authors efforts at clarity and maintain my assessment that this is a good paper and should be accepted.

---

### Meta-Review · Area_Chair_2Sjp · 2024-12-19

**Metareview:**

This paper is truly in the borderline with varied reviews. The reviewers liked the idea of automated solutions in ILP, something that is needed sorely if the field has to be sustained in the current era of AI. From that perspective, this is an important paper with an important dataset for understanding the limitations of ILP methods.

The unfavorable opinions stem from the fact that the paper did not properly motivate the use of high-arity predicates when simple decompositions seem to work well in practice. The notion was that the paper created a circular argument by creating problems that are beyond the current systems and then claiming to solve them. It is unclear if there are much practical use of this (again, this goes back to lack of motivation).

From my own reading of the paper and assessment, I find the data set and the solution interesting. The paper can improve its motivation better and I hope the authors consider the reviews seriously when submitting the final version.

**Additional Comments On Reviewer Discussion:**

There was reasonable discussion with the reviewer with negative score keeping their score (I am not sure I fully agree with the decision to keep the score after the authors had addressed several of their claims).

---

### Decision · Program_Chairs · 2025-01-22

Accept (Poster)